# The geometry of robustness in spiking neural networks

**Nuno Calaim[1†], Florian A Dehmelt[1,2†], Pedro J Gonçalves[3,4,5†], Christian K Machens[1]\***

[1]Champalimaud Neuroscience Programme, Champalimaud Foundation, Lisbon, Portugal; [2]Centre for Integrative Neuroscience, Tübingen University Hospital, Tübingen, Germany; [3]Center of Advanced European Studies and Research (CAESAR), Bonn, Germany; [4]Computational Neuroengineering, Department of Electrical and Computer Engineering, Technical University of Munich, Munich, Germany; [5]Machine Learning in Science, Excellence Cluster 'Machine Learning', Tübingen University, Tübingen, Germany

**Abstract** Neural systems are remarkably robust against various perturbations, a phenomenon that still requires a clear explanation. Here, we graphically illustrate how neural networks can become robust. We study spiking networks that generate low-dimensional representations, and we show that the neurons' subthreshold voltages are confined to a convex region in a lower-dimensional voltage subspace, which we call a 'bounding box'. Any changes in network parameters (such as number of neurons, dimensionality of inputs, firing thresholds, synaptic weights, or transmission delays) can all be understood as deformations of this bounding box. Using these insights, we show that functionality is preserved as long as perturbations do not destroy the integrity of the bounding box. We suggest that the principles underlying robustness in these networks — low-dimensional representations, heterogeneity of tuning, and precise negative feedback — may be key to understanding the robustness of neural systems at the circuit level.

**\*For correspondence:**
christian.machens@neuro.
fchampalimaud.org

[†]These authors contributed
equally to this work

**Competing interest:** The authors declare that no competing interests exist.

## Editor's evaluation

The article introduces a geometrical interpretation for the dynamics and function of certain spiking networks, based on the earlier work of Machens and Deneve. Given that spiking networks are notoriously hard to understand, the approach could prove useful for many computational neuroscientists. Here, that visualization tool serves to assess how fragile the network is to perturbation of its parameters, such as neuronal death, or spurious noise in excitation and inhibition.

## Introduction

The ability to maintain functionality despite perturbations is one of the defining properties of biological systems, from molecular signaling pathways to whole ecosystems (*Csete and Doyle, 2002*; *Kitano, 2004*; *Whitacre, 2012*; *Félix and Barkoulas, 2015*). Neural systems likewise withstand a certain amount of damage or external disturbances, which is evident from lesion studies or neurodegenerative diseases (*Morrison and Hof, 1997*; *Bredesen et al., 2006*; *Palop et al., 2006*), as well as from perturbation experiments (*Wolff and Ölveczky, 2018*; *Li et al., 2016*; *Trouche et al., 2016*; *Fetsch et al., 2018*). However, the mechanisms that underlie this robustness are not entirely clear. Indeed, most models of neural networks, when faced with partial damage, lose their functionality quite rapidly (*Figure 1A–C*; *Seung et al., 2000*; *Koulakov et al., 2002*; *Li et al., 2016*). Beyond its

biological interest, understanding the robustness of neural systems is also crucial for the correct interpretation of experiments that seek to manipulate neural circuits (*Wolff and Ölveczky, 2018*).

Robustness has sometimes been attributed to various single-cell mechanisms, such as those that stabilize the dynamics of the stomatogastric ganglion of crustaceans against temperature fluctuations (*O'Leary and Marder, 2016*; *Haddad and Marder, 2018*), or the oculomotor integrator against instabilities in positive feedback loops (*Koulakov et al., 2002*; *Goldman et al., 2003*). On the circuit-level, robustness has been tied to the excitatory-inhibitory (EI) balance of synaptic inputs, either by linking such balance with the efficiency of neural population codes (*Boerlin et al., 2013*; *Bourdoukan et al., 2012*; *Barrett et al., 2013*), or by using it as a corrective feedback for integrator models (*Lim and Goldman, 2013*). The corresponding spiking networks can maintain functional representations of their inputs by re-balancing when faced with perturbations such as neuron loss (*Barrett et al., 2016*; *Lim and Goldman, 2013*). *Figure 1D* illustrates such a robust, spiking network model.

Here, we illustrate how circuits can be made robust through simple geometric insights that tie the low-dimensional representations found in many population recordings (*Saxena and Cunningham, 2019*; *Keemink and Machens, 2019*; *Vyas et al., 2020*) to the biophysics of neurons, such as their voltages, thresholds, and synaptic inputs. We therefore provide a principled theory on how networks may have become robust to the many perturbations encountered in nature. We use this theory to illustrate two effects. First, we show that the resulting robustness mechanisms include the balanced regime, but are not limited to it. Indeed, networks can be robust without exhibiting any EI balance. Second, we predict a surprising asymmetry to perturbations: we find that robust networks are insensitive to broad inhibitory perturbations, yet quite sensitive to small excitatory perturbations, even if the latter are restricted to single neurons in large networks. This heightened sensitivity may explain the ability of animals to recognize exceedingly small, excitatory perturbations (*Houweling and Brecht, 2008*; *Huber et al., 2008*; *Dalgleish et al., 2020*).

Beyond questions of robustness, our work also provides a new way of thinking about spiking networks, which complements and extends classical approaches such as mean-field or attractor dynamics. To simplify our exposition, we focus on generic networks of integrate-and-fire neurons, rather than modeling a specific system. Consequently, we ignore part of the biological complexity (e.g. Dale's law, more complex computations or dynamics), and defer explanations on how the framework may generalize to more realistic network models to the discussion and the methods.

## Results

Our first assumption is that neural networks generate low-dimensional representations of sensory or motor signals, which can be extracted from the spike trains of a neural population through filtering and summation. Here, 'low-dimensional' simply means that the number of signals (or dimensions) represented is far less than the number of neurons in a circuit, so that there exists a certain amount of redundancy. Such redundant representations have been observed in many brain circuits (*Saxena and Cunningham, 2019*; *Keemink and Machens, 2019*), and are an integral part of most mid- to large-scale network models (*Vogels et al., 2005*; *Eliasmith and Anderson, 2004*; *Barak, 2017*). However, they do not per se guarantee robustness as shown in an example network in *Figure 1C*.

### Passive redundancy

A classical example of a redundant, but non-robust representation is a sensory layer of $N$ independent neurons acting as feature detectors. Here, each neuron receives the same $M$ time-varying input signals, $(x_1(t), x_2(t), \ldots, x_M(t)) = \mathbf{x}(t)$. Each signal is weighted at a neuron's synapse, and the resulting synaptic currents are then summed in the neuron's soma. If the synaptic weights are chosen differently for different neurons, the population generates a distributed code, whose redundancy we define as the number of neurons per signal dimension, or $\rho = N/M$. We will call this redundancy 'passive' as each neuron fires completely independent of what other neurons are doing.

While actual sensory systems are obviously more complex, this layer of independent feature detectors still serves as a useful baseline. For instance, in such a layer, perturbing a set of neurons by exciting or inhibiting them will have an effect on the representation that is directly proportional to the number of neurons perturbed. Passive redundancy therefore leads to a gradual decline of functionality (*Figure 1B*) or a gradual response of a system to any perturbation.

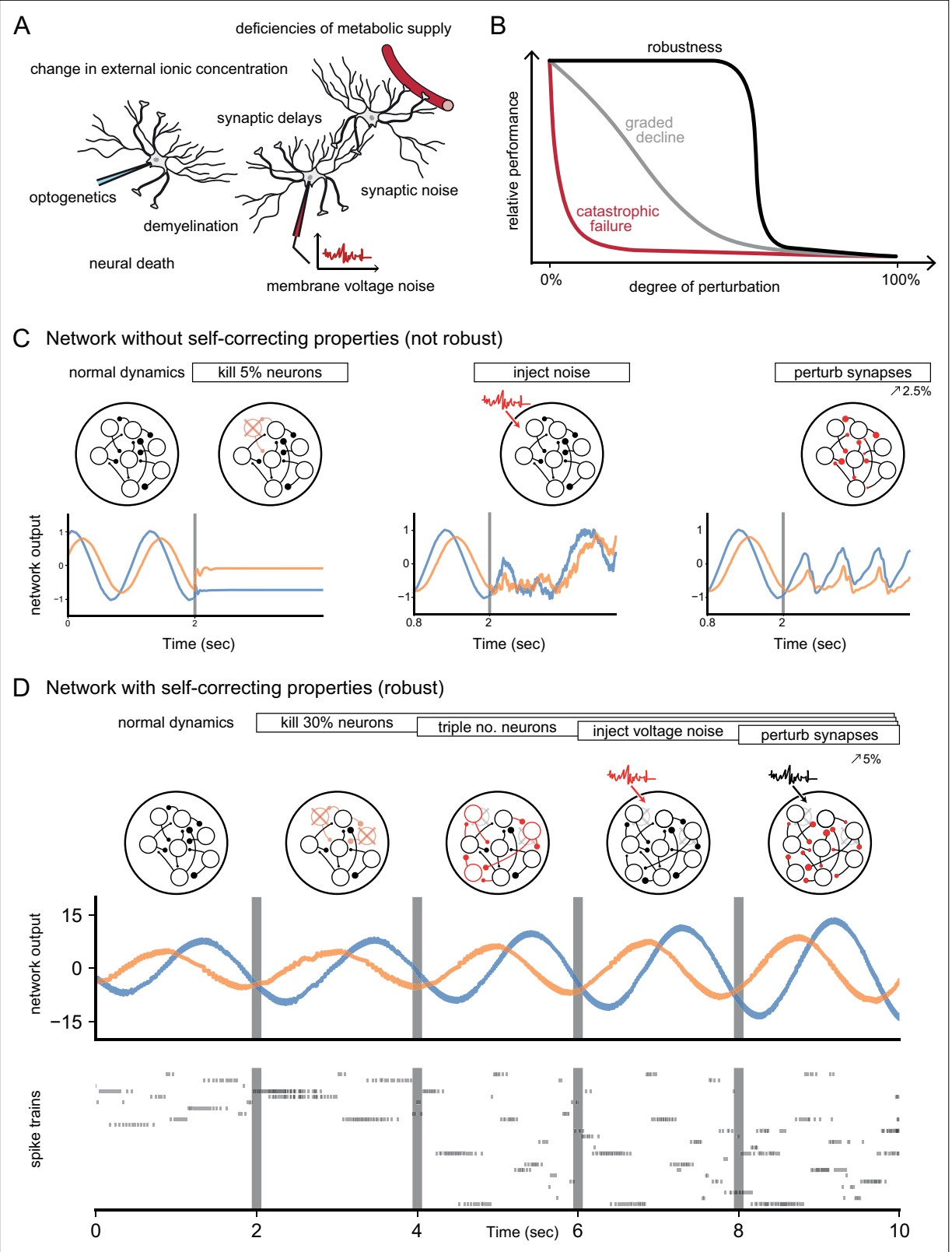

**Figure 1.** Neural systems are robust against a variety of perturbations. (**A**) Biological neural networks operate under multiple perturbations. (**B**) The degree of robustness of a system can fall into three regimes: 1. Catastrophic failure (red), when small changes in the conditions lead to quick loss of function for the system. 2. Gradual degradation (gray), when the system's performance is gradually lost when departing from optimal conditions. 3. Robust operation (black), when the network is able to maintain its function for a range of perturbations. (**C**) Most rate- and spike-based network models

*Figure 1 continued on next page*

*Figure 1 continued*

fail to withstand even small perturbations. Shown here is a rate network (composed of $N = 1000$ neurons) trained with FORCE-learning to generate a two-dimensional oscillation (*Sussillo and Abbott, 2009*). The performance of the trained network declines rapidly when exposed to a diverse set of perturbations. Other learning schemes yield similar results. (**D**) By contrast, a network in which neurons coordinate their firing to correct any errors is robust to several, even cumulative perturbations. Shown here is a spiking network composed of initially $N = 10$ neurons, designed to generate a two-dimensional oscillation (*Boerlin et al., 2013*). *Top*: Schematic of the various perturbations. Vertical lines indicate when a new perturbation is added. The diffusion coefficient of the injected voltage noise is more than 5% of the neuronal threshold magnitude. The perturbation of all synaptic weights is random and limited to 5%. *Middle*: Two-dimensional output, as decoded from the network activity. *Bottom*: Raster plot of the network's spike trains.

## Autoencoder with low-dimensional readouts

To create robustness to perturbations, neurons cannot act independently, but rather need to coordinate their firing. We will now consider an example network that is generating a sensory representation at the population level, such that this representation is optimal with respect to a given linear readout (*Figure 2A*). We will focus on this simple scenario in order to highlight the mechanisms that endow networks with robustness. In the Discussion and Materials and methods, we will point out how to transfer these insights to more general networks.

Just as above, we consider a network of $N$ neurons which receive an $M$-dimensional vector of input signals, $\mathbf{x}(t)$. The task of the network is to be an autoencoder, that is, to generate spike trains such that the input signals can be read out by a downstream area. We assume a linear readout, which filters each spike train with an exponential filter, similar to the postsynaptic potentials generated in a single synapse. Then, the filtered spike trains are weighted and summed, similar to the passive summation in a dendritic tree. Formally, we write

$$\hat{\mathbf{x}}(t) = \sum_{k=1}^{N} \mathbf{D}_k r_k(t), \tag{1}$$

where $r_k(t)$ is the filtered spike train of the $k$-th neuron, $N$ is the number of neurons, $\hat{\mathbf{x}}(t) = \left(\hat{x}_1(t), \hat{x}_2(t), \ldots, \hat{x}_M(t)\right)$ is the vector of readouts, distinguished from the input signals by a hat, and $\mathbf{D}_k = (D_{1k}, D_{2k}, \ldots, D_{Mk})$ is the decoding vector of the $k$-th neuron, whose individual elements contain the respective decoding weights.

We can depict the geometrical consequences of this decoding mechanism by imagining a network of five neurons that is encoding two signals. At a given point in time, we can illustrate both the input signals and the readout produced by the network as two points in signal space (*Figure 2B*, black cross and gray dot). Now let us imagine that one of the neurons, say neuron $i$, spikes. When that happens, the spike causes a jump in its filtered output spike train. In turn, and according to *Equation 1*, the vector of readouts, $\hat{\mathbf{x}}$, jumps in the direction of the decoding vector, $\mathbf{D}_i$, as illustrated in *Figure 2B*. Since the direction and magnitude of this jump are determined by the fixed readout weights, they are independent of the past spike history or the current values of the readouts. After this jump, and until another neuron fires, all components of the readout will decay. Geometrically, this decay corresponds to a movement of the readout towards the origin of the coordinate system.

## Coordinated redundancy and the error bounding box

We furthermore assume that a neuron spikes only when its spike moves the readout closer to the desired signal, $\mathbf{x}$. For each neuron, this spike rule divides the whole signal space into two regions: a 'spike' half-space where the readout error decreases if the neuron spikes, and a 'no-spike' half-space where the readout error increases if the neuron spikes (*Figure 2B*). The boundary between these two half spaces is the neuron's spiking threshold, as seen in signal space. Consequently, the neuron's voltage, $V_i$, must be at threshold, $T_i$, whenever the readout reaches this boundary, and the voltage must be below or above threshold on either side of it. We therefore identify the neuron's voltage with the geometric projection of the readout error onto the decoding vector of the neuron,

$$V_i = \mathbf{D}_i^{\top} (\mathbf{x} - \hat{\mathbf{x}}), \tag{2}$$

where, without loss of generality, we have assumed that $\mathbf{D}_i$ has unit length (see Materials and methods, 'Coordinated spiking and the bounding box'). The effect of this definition is illustrated in *Figure 2E*, where the voltage increases or decreases with distance to the boundary. Accordingly, the voltage measures part of the error, given here by the distance of the readout to the neuron's boundary.

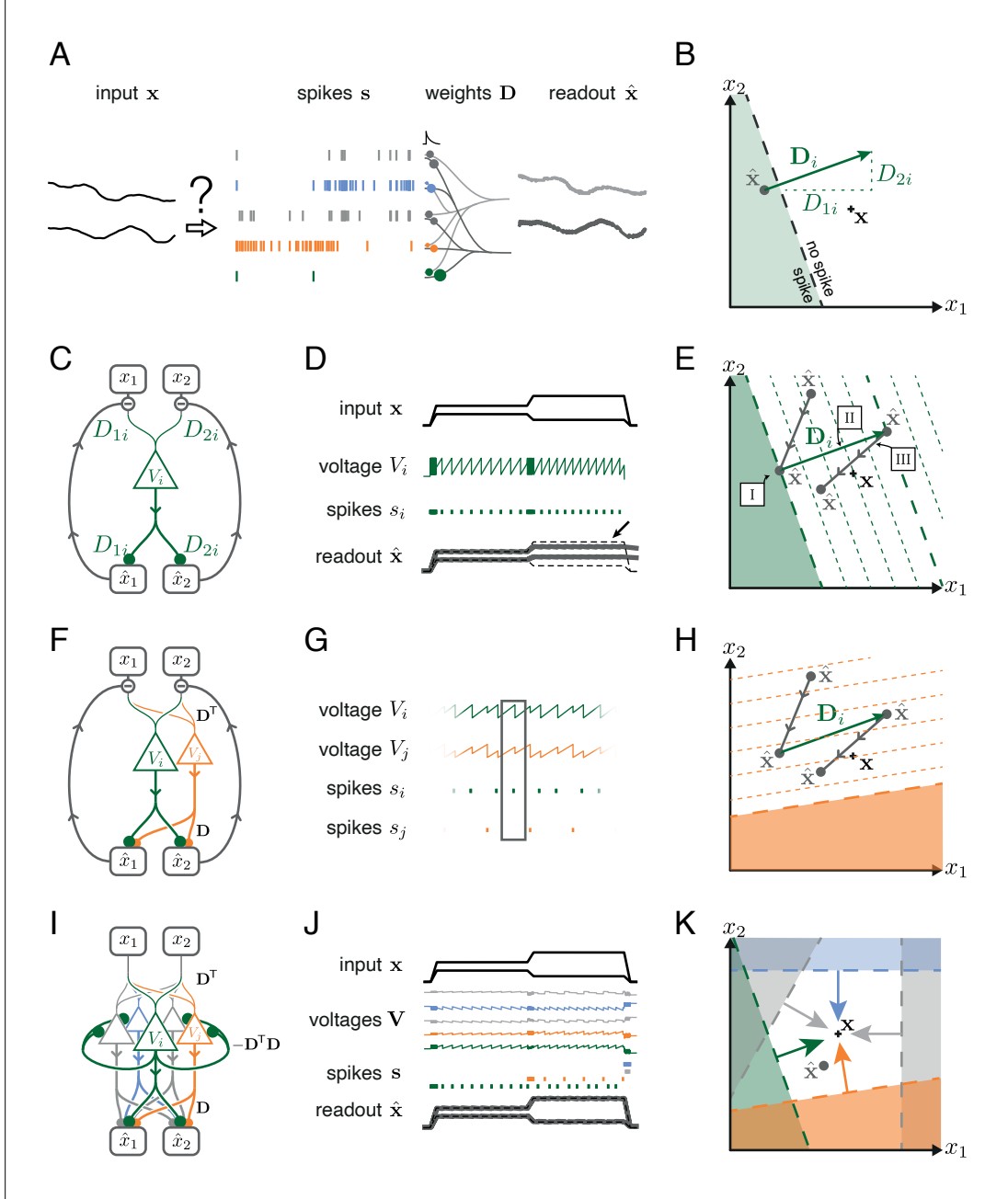

**Figure 2.** Toy example of a network with coordinated redundancy ($M = 2$ inputs and $N = 5$ neurons). (**A**) The task of the network is to encode two input signals (black) into spike trains (colored), such that the two signals can be reconstructed by filtering the spike trains postsynaptically (with an exponential kernel), and weighting and summing them with a decoding weight matrix $\mathbf{D}$. (**B**) A neuron's spike moves the readout in a direction determined by its vector of decoding weights. When the readout is in the 'spike' region, then a spike from the neuron decreases the signal reconstruction error. Outside of this region ('no spike' region), a spike would increase the error and therefore be detrimental. (**C**) Schematic diagram of one neuron. Inputs arrive from the top. The neuron's voltage measures the difference between the weighted input signals and weighted readouts. (**D**) Simulation of one neuron tracking the inputs. As one neuron can only monitor a single error direction, the reconstructed signal does not correctly track the full two-dimensional signal (arrow). (**E**) Voltage of the neuron (green) and example trajectory of the readout (gray). The dashed green lines correspond to points in space for which neuron    has the same voltage (voltage isoclines). The example trajectory shows the decay of the readout until the threshold is reached (I), the jump caused by the firing of a spike (II), and the subsequent decay (III). (**F**) Same as C, but considering two different neurons. (**G**) Voltages and spikes of the two neurons. (**H**) Voltage of the orange neuron during the same example trajectory as in E. Note that the neuron's voltage jumps during the firing of the spike from the green neuron. (**I**) The negative feedback of the readout can be equivalently implemented through lateral connectivity with a weight matrix $\mathbf{\Omega} = -\mathbf{D}^{\mathsf{T}}\mathbf{D}$. (**J**) Simulation of five neurons tracking the inputs. Neurons coordinate their spiking such that the readout units can reconstruct the input signals up to a precision given by the size of the error bounding box. (**K**) The network creates an error bounding box around $\mathbf{x}$. Whenever the network estimate $\hat{\mathbf{x}}$ hits an edge of the box, the corresponding neuron emits a spike pushing the readout estimate back inside the box (colored arrows).

In addition to its functional interpretation, the voltage equation has a simple biophysical interpretation, as illustrated in *Figure 2C*. Here, the two input signals, $x_1$ and $x_2$, get weighted by two synaptic weights, $D_{1i}$ and $D_{2i}$, leading to two postsynaptic voltages that are then summed in the dendritic tree of neuron $i$. At the same time, the two readouts, $\hat{x}_1$ and $\hat{x}_2$, are fed back into the neuron via two exactly opposite synaptic weights, $-D_{1i}$ and $-D_{2i}$, thereby giving rise to the required subtraction. As a consequence, the neuron's voltage becomes the projection of the readout error, as prescribed above. When the neuron's voltage reaches the voltage threshold, $T_i$, the neuron fires a spike, which changes the readout, $\hat{\mathbf{x}}$. In turn, this change is fed back into the neuron's dendritic tree and leads to an effective reset of the voltage after a spike, as shown in *Figure 2D*.

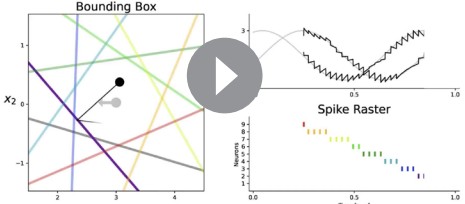

Normal operation of spiking network

**Video 1.** Normal operation of a network with two- or three-dimensional inputs. Shown are an animation of the bounding box dynamics, the input signal and readout, and the spike trains produced by the network. https://elifesciences.org/articles/73276/figures#video1

One neuron alone can only improve the readout along one specific direction in signal space and thus cannot correct the readout for all possible input signals (*Figure 2D*, arrow). To properly bound the error, we therefore need several neurons, and each neuron needs to mix the input signals in different ways. As a consequence, the error will be corrected along different directions in signal space. A second neuron, say neuron $j$, is added in *Figure 2F–H*. Following the logic above, its voltage is given by $V_j = \mathbf{D}_j^\top (\mathbf{x} - \hat{\mathbf{x}})$, and the respective voltage isoclines are shown in *Figure 2H*. We see that the voltage of neuron $j$ jumps when neuron $i$ spikes. Mathematically, the size of this jump is simply given by the dot product of the two decoding vectors, $\mathbf{D}_j^\top \mathbf{D}_i$. Biophysically, such a jump could be caused by negative feedback through the readout units, but it could also arise through a direct synaptic connection between the two neurons, in which case $\Omega_{ji} = -\mathbf{D}_j^\top \mathbf{D}_i$ corresponds to the synaptic weight from neuron $i$ to neuron $j$.

Finally, if we add three more neurons, and give them different sets of decoding weights, the network as a whole can restrict the readout to a bounded region in signal space (a polygon in two dimensions), as shown in *Figure 2I–K*. We will call this bounded region the 'error bounding box' or simply the 'bounding box.' Its overall size determines the error tolerance of the network. To highlight the structure of this network, we can change *Equation 2* by inserting the definition of the readout, *Equation 1*, to obtain

$$V_i = \mathbf{D}_i^\top \mathbf{x} - \sum_{k=1}^N \mathbf{D}_i^\top \mathbf{D}_k r_k. \tag{3}$$

Here, the term $\Omega_{ik} = -\mathbf{D}_i^\top \mathbf{D}_k$ can be interpreted as a lateral connection between neurons $i$ and $k$ in the network (*Figure 2I*). The diagonal elements of the respective connectivity matrix, $\Omega_{ii}$, can be interpreted as the hyperpolarization of the membrane voltage following a spike. While the connectivity of the network is symmetric, this assumption can be relaxed (see Materials and methods, 'Generalization of the bounding box II'). The connectivity term shows that information about the readout can be relayed through lateral connections and self-resets (*Figure 2I*), rather than through explicit negative feedback from a downstream layer. In either case, the feedback causes neurons to coordinate their firing. We will refer to this mechanism as 'coordinated spike coding' (*Boahen, 2017*) or 'coordinated redundancy'.

As shown previously (*Bourdoukan et al., 2012*; *Boerlin et al., 2013*), the temporal derivative of the above equation yields a network of current-based, leaky integrate-and-fire neurons (see Materials and methods, 'Coordinated spiking and the bounding box'). We emphasize that there are two distinct situations that cause neurons to emit spikes. First, the readout always leaks towards the origin, and when it hits one of the boundaries, the appropriate neuron fires and resets the readout into the bounding box. Second, any change in the input signal, $\mathbf{x}$, causes a shift of the entire bounding box, since the signal is always at the centre of the box. A sudden shift may therefore cause the readout to fall outside of the box, in which case neurons whose boundaries have been crossed will fire to get the readout back into the box.

When the signal dimensionality is increased to $M = 3$, the thresholds of the neurons become planes, and the bounding box is determined by the intersection of all planes, thereby becoming a three-dimensional object such as a soccer ball. We strongly recommend to view *Video 1* for an animation of the operation of the network in two and three dimensions, which highlights the relation between the bounding box and the resulting spike trains produced by the network.

## The bounding box limits the coding error

The maximum coding error is limited by the size of the error bounding box, simply because the readout cannot deviate from the signal beyond the borders of the box. The size of the box is determined by the neurons' thresholds. For simplicity, we will assume that all thresholds are identical, which could be regulated through homeostatic mechanisms (*Turrigiano, 2012*). The more general scenario is explained in Materials and methods, 'Generalization of the bounding box I'.

Beyond changing the maximum allowable coding error, the size of the error bounding box also influences the resulting code in more subtle ways. First, the coding error can be split into a systematic bias and random fluctuations (see *Appendix 1—figure 1A*). As the box becomes wider, the systematic bias increases. This bias can be largely eliminated by rescaling the readouts with a constant factor. We will sometimes use this corrected readout (see Materials and methods, 'Readout biases and corrections'), but note that the corrected readout is not confined to stay within the bounding box. Second, if the box becomes very narrow, the readout can eventually jump beyond the boundary of the opposite side, thereby crossing the threshold(s) of oppositely tuned neurons (see also *Appendix 1—figure 1B–D*). By default, we will assume that the bounding box is sufficiently wide to avoid this effect.

Finally, while the bounding box may seem like a fairly abstract construction, we note that it also has a simple, physical manifestation. Since the neurons' voltages are constrained to live in an $M$-dimensional subspace, a constraint given by the input dimensionality, the error bounding box delineates the borders of this voltage subspace, which is illustrated in *Appendix 1—figure 2*.

## Robustness to inhibitory, sensitivity to excitatory perturbations

We will now study how the network reacts to perturbations by contrasting two example networks at opposite ends of a spectrum. In the first network, neurons are independent. For a two-dimensional signal, we obtain this scenario when the bounding box consists of four neurons forming a square (*Figure 3A*, left), in which case neighbouring decoding vectors are orthogonal to each other, and their recurrent connections disappear ($\mathbf{D}_i^\top \mathbf{D}_k = 0$, see also Materials and methods, 'Generalization of the bounding box III'). The second network consists of $N = 21$ randomly tuned neurons with equidistant thresholds, in which case the bounding box approximates the shape of a circle (*Figure 3B*, left).

The first perturbation we consider is the death of a single neuron. Throughout an organism's life, cells, including neurons, can undergo the process of cell death or apoptosis if they are damaged or unfit, as may happen in diseased states (*Moreno et al., 2015*; *Morrison and Hof, 1997*; *Bredesen et al., 2006*; *Coelho et al., 2018*). Geometrically, the death of a neuron is equivalent to the removal of its corresponding face from the bounding box (*Figure 3A and B*, and *Video 2*). When the bounding box is breached on one side, the readout can no longer contain changes of the input signal along the open direction. This is precisely what happens in the case without redundancy (*Figure 3A*, right). In contrast, with coordinated redundancy, the removal of a single neuron has an almost imperceptible impact on the shape of the bounding box (*Figure 3B*, right). Consequently, the coding error remains bounded with essentially unchanged precision. The bounding box provides therefore a straightforward and intuitive explanation for the robustness against neuron loss observed in these spiking networks (*Barrett et al., 2016*; *Boerlin et al., 2013*).

The second perturbation we consider is a change in the excitability of one neuron. Such a change could come about through an experimentally injected current, for example, via patch clamp or optogenetics, or because of intrinsic plasticity. In either case, a change in excitability is equivalent to a change in the neuron's threshold (see Materials and methods, 'Perturbations'). Within the bounding box picture, an inhibitory perturbation or decrease in excitability leads to an outward shift of the neuron's threshold, and an excitatory perturbation or increase in excitability leads to an inward shift (*Figure 3C and D*). Without redundancy, the bounding box expands or shrinks, respectively (*Figure 3C*). At first sight, changing the box size increases or decreases the maximum error of the readout. More subtly, however, it also introduces a bias in the corrected average readout (*Figure 3C*,

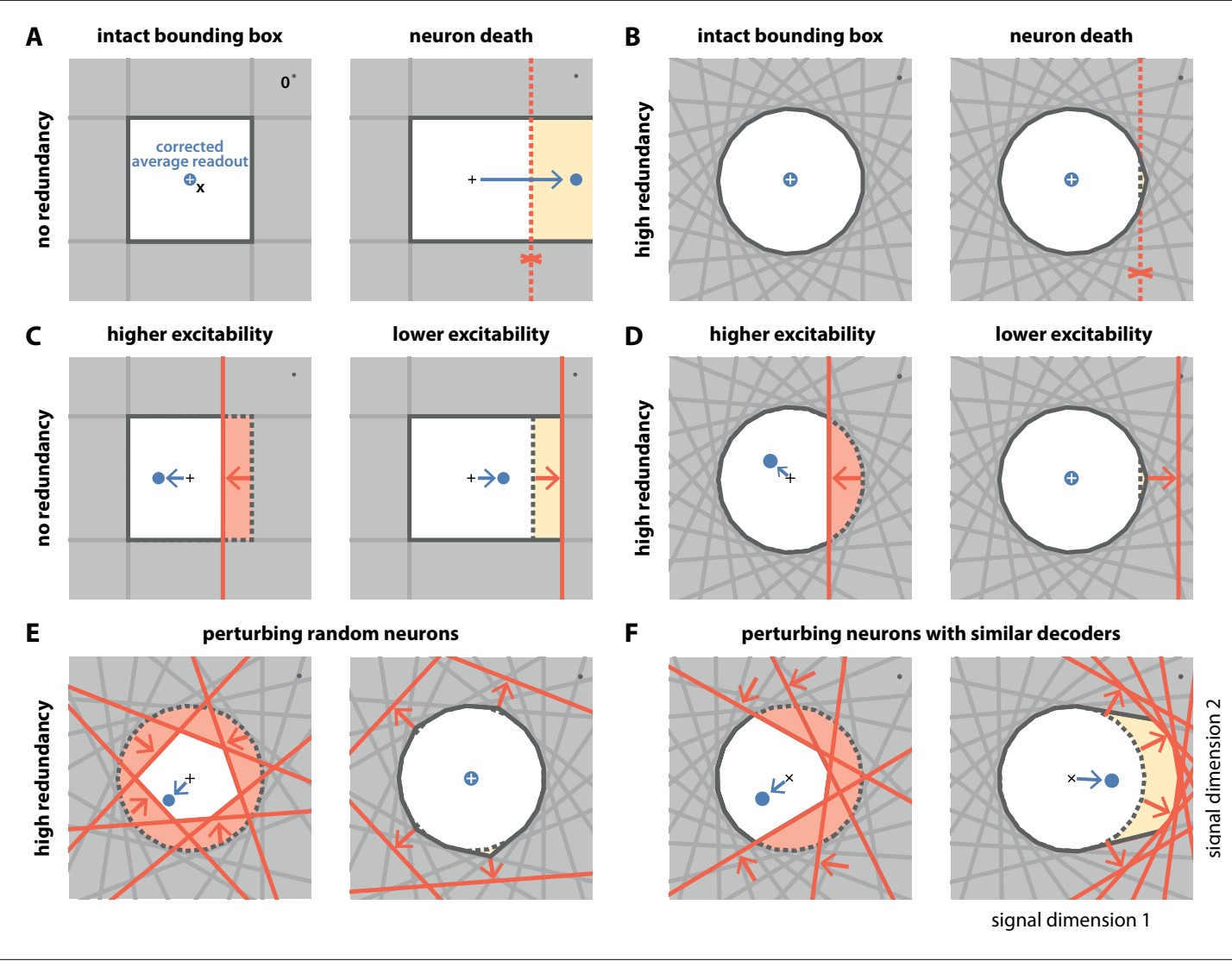

**Figure 3.** Geometry of perturbations. (**A**, left) A network of four independent neurons without redundancy. The bias-corrected, average readout (blue circle) is identical to the input signal (white cross). (**A**, right) When one neuron dies, the bounding box opens to one side, and the readout is no longer contained in the respective direction. In turn, the time-averaged readout moves to the right (blue dot) for the applied input signal (cross). (**B**) In a network of $N = 21$ neurons with coordinated redundancy (left), neural death has almost no impact on bounding box shape and decoding error (right). (**C**) In the network without redundancy, an increase (left) or decrease (right) in the excitability of one neuron changes the size of the box, but the box remains bounded on all sides. The corrected readout shifts slightly in both cases. (**D**) In the same network, increased excitability (left) has the same effect as in a non-redundant network, unless the box is reduced enough to trigger ping-pong (*Appendix 1—figure 1C–D*). Decreased excitability (right) has virtually no effect. (**E,F**) If several neurons are perturbed simultaneously, their relative decoder tuning determines the effect. (E, left) Increasing the excitability of multiple, randomly chosen neurons has the same qualitative effect as the perturbation of a single neuron. However, in this case, the smaller box size pushes the corrected readout away from the origin of the signal space. (E, right) Decreasing the excitability of multiple neurons has little effect. (**F**) If neurons with similar tuning are targeted, both higher (left) and lower (right) excitability significantly alter the box shape and alter the corrected readout.

arrows). With coordinated redundancy, inhibitory and excitatory perturbations do not have opposite effects. Whereas an excitatory perturbation has an effect equivalent in size to a non-redundant system, as the threshold that is pushed inwards shrinks the box (*Figure 3D*, left), an inhibitory perturbation does not affect the system at all, because the outward shift of the threshold is compensated by the presence of other neurons (*Figure 3D*, right, see also *Video 2*). We note that a strong excitatory perturbation could cause the readout to move beyond the boundary of the opposite side,

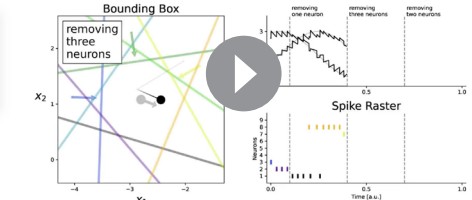

**Video 2.** Operation of a network with two-dimensional inputs, under different perturbations, namely neural death, voltage noise, change in voltage resets, synaptic perturbations, delays, and inhibitory and excitatory optogenetic perturbations. Shown are the bounding box, input signals and readouts, and the spike trains produced by the network.

https://elifesciences.org/articles/73276/figures#video2

thereby leading to the undesirable 'ping-pong' effect (*Appendix 1—figure 1*). This effect can be avoided if the bounding box is sufficiently wide.

In most experimental settings, one will perturb many neurons at once. We illustrate the effects of two such perturbations in *Figure 3E and F*. In *Figure 3E*, left, we excite six randomly selected neurons so that their boundaries are pushed inwards. As a consequence, the bounding box shrinks equally from all sides (*Figure 3E*, right). Just as in *Figure 3D*, left panel, the respective inward shifts cause (input-signal-dependent) biases in the corrected readout. In contrast, when we randomly inhibit neurons as in *Figure 3E*, right, the respective boundaries are pushed outwards. The bounding box barely changes, as the remaining neurons contain the readout error with essentially unchanged precision, and the system remains functional. In *Figure 3F*, we excite and inhibit neurons with similar tuning. In this case, the inhibitory perturbation is so large that there are no longer any neurons that can compensate. As a consequence, the bounding box expands in the perturbed direction and the corrected readout becomes biased, even for inhibitory perturbations.

In summary, we observe that coordinated redundancy endows the system with robustness against perturbations that act inhibitorily on the neurons, such as neuron death or increases in spiking thresholds. The function of the system remains intact until the bounding box breaks open. However, coordinated redundancy also makes the system highly sensitive towards any excitatory perturbations or decreases in spiking thresholds. Indeed, perturbing only a single neuron is sufficient to generate a change in the readout. These results contrast with passively redundant systems, whose representation changes gradually with either excitatory or inhibitory perturbations.

We note that there is circumstantial evidence that cortical systems are indeed highly sensitive to excitatory perturbations, and potentially less sensitive to inhibitory perturbations. For instance, animals can detect excitatory currents injected into a few pyramidal cells in somatosensory cortex (*Houweling and Brecht, 2008*; *Huber et al., 2008*; *Dalgleish et al., 2020*). To our knowledge, no study has shown that cortical systems are similarly sensitive to inhibitory perturbations in one or a few neurons. Rather, several studies have shown that neural systems can compensate for inhibitory perturbations in a sizeable fraction of pyramidal cells (*Li et al., 2016*; *Fetsch et al., 2018*; *Trouche et al., 2016*).

## The neurophysiological signatures of perturbations

Besides insights into a system's functionality, the bounding box also allows us to see immediately how perturbations affect the firing of the unperturbed neurons. For instance, when we excite a single neuron, its threshold moves inwards (*Figure 3D*) and occludes the thresholds of the neighboring neurons, that is, neurons with similar selectivity. Since the readout can no longer reach these neurons, they stop firing, as shown in a simulation of the network in *Figure 4A*. Conversely, if we inhibit one or more neurons, their thresholds become hidden and they no longer participate in containing the readout (*Figure 3D–F*). As a consequence, the surrounding neurons have to pick up the bill and fire more, so that their firing rates will increase, as shown in *Figure 4A*. Biophysically, these effects are of course mediated by an increase or decrease of lateral or recurrent inhibition. However, the bounding box provides a simple visualization of the effect and its purpose.

The bounding box also allows us to visualize the relation between EI balance and robustness. We can again study two extreme examples, as illustrated in *Figure 4B and C*. Here, both bounding boxes are intact and contain the readout. The box in *Figure 4B* has low redundancy ($N = 5$ neurons for $M = 2$ signals), whereas the box in *Figure 4C* has high redundancy ($N = 100$ neurons for $M = 2$ signals). In

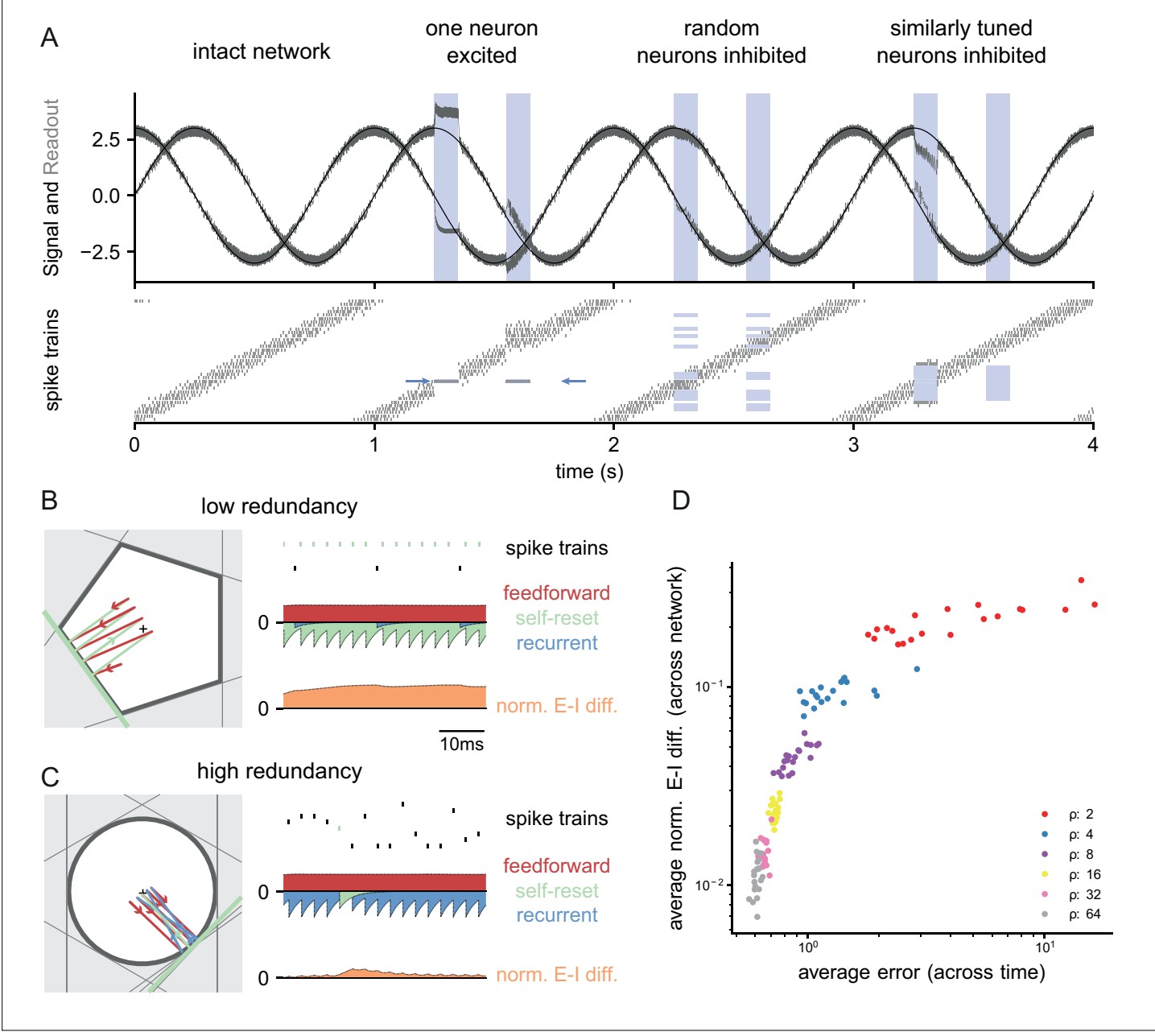

**Figure 4.** Neurophysiological signatures of perturbations. (**A**) Asymmetric effects of excitatory and inhibitory perturbations. Shown are the two input signals (black lines), corrected readouts (gray lines), and spike trains (raster plot) during different perturbations (blue boxes). The excitation of a single neuron (blue arrows) is sufficient to perturb the readout. In contrast, the network remains fully functional when a random subset of neurons is inhibited. Here, the remaining neurons compensate for the loss by firing more spikes. However, a bias occurs when a sufficiently large set of similarly tuned and active neurons are inhibited. Here, the compensation of neighboring neurons is not sufficient to contain the error. (**B**) Network with low redundancy ($N = 5$ neurons and $M = 2$ signals). The left panel illustrates the bounding box, and the trajectory of the readouts, color-coded by changes due to the neuron's spikes (green), the feedforward inputs (red) and the recurrent inputs (blue). The right panel shows the spikes of the network (top, green neuron highlighted), the input currents into the green neuron as a function of time (middle), and the difference between the synaptic excitatory and inhibitory input currents (bottom). In this example, the currents are dominated by excitatory feedforward inputs and self-reset currents, thereby causing a positive E-I difference. (**C**) Network with high redundancy ($N = 100$ neurons and $M = 2$ signals). Same format as (**B**). In this example, the feedforward currents are balanced by recurrent inputs of equal strength, but opposite sign. The recurrent inputs here replace the self-reset currents and correspond to input spikes of other neurons that have hit their respective thresholds, and take care of the coding error. As a consequence, the green neuron is tightly balanced. (**D**) Average normalized E-I difference and average coding error as a function of the redundancy $\rho$ (color-coded). The average coding error remains low even in a regime where substantial parts of the population are already imbalanced.

turn, we can visualize the synaptic current balance in these boxes by highlighting how the movements of the readout give rise to excitatory or inhibitory currents.

In the low redundancy case, the readout initially decays, and thereby moves towards the threshold of one of the neurons (shown in green). A movement towards threshold corresponds to a depolarization of the respective membrane potential and therefore an excitatory (or dis-inhibitory) current, here illustrated in red. The respective neuron spikes, and the readout jumps away from the threshold, which is mediated by a hyperpolarizing self-reset current, here illustrated in green. Since the excitatory drive is cancelled by self-resets following spiking, the synaptic inputs are not balanced, and the neuron fires spikes in a relatively regular rhythm (*Figure 4B*, right panel, see Materials and methods, 'Excitation-inhibition balance').

The situation is quite different for the high-redundancy network (*Figure 4C*). Here, the readout decays towards the thresholds of multiple neurons (some of which are highlighted with gray lines), but only one of these neurons will fire. When that happens, all neurons in the vicinity immediately receive inhibitory currents that signal the concomitant change in the readout. These inhibitory currents thereby cancel the excitatory feedforward drive, and the respective neurons experience a tight EI balance, leading to sparse and irregular spike trains (*Figure 4C*, left panel). We note that this tight balance only holds in the neurons that are sufficiently close to the neuron whose threshold is crossed. Neurons further away will experience only small current fluctuations, or will, on average, be hyperpolarized.

As a result, we see that EI balance occurs in networks that are sufficiently redundant, but not in networks with no or low redundancy. Nonetheless, even networks with low redundancy have a measure of robustness: for instance, the network in *Figure 4B* is robust against the loss of one neuron. While previous work has suggested that networks recover functionality when perturbed by dynamically re-establishing EI balance (*Lim and Goldman, 2013*; *Boerlin et al., 2013*; *Barrett et al., 2016*), our considerations here show that robustness extends beyond the regime of EI balance. *Figure 4D* illustrates this result by contrasting the performance and balance of networks as a function of their redundancy.

## Scaling up

While the simple toy networks we have studied so far are useful for illustration and intuition, biological neural networks, and especially cortical networks, consist of thousands of neurons that are thought

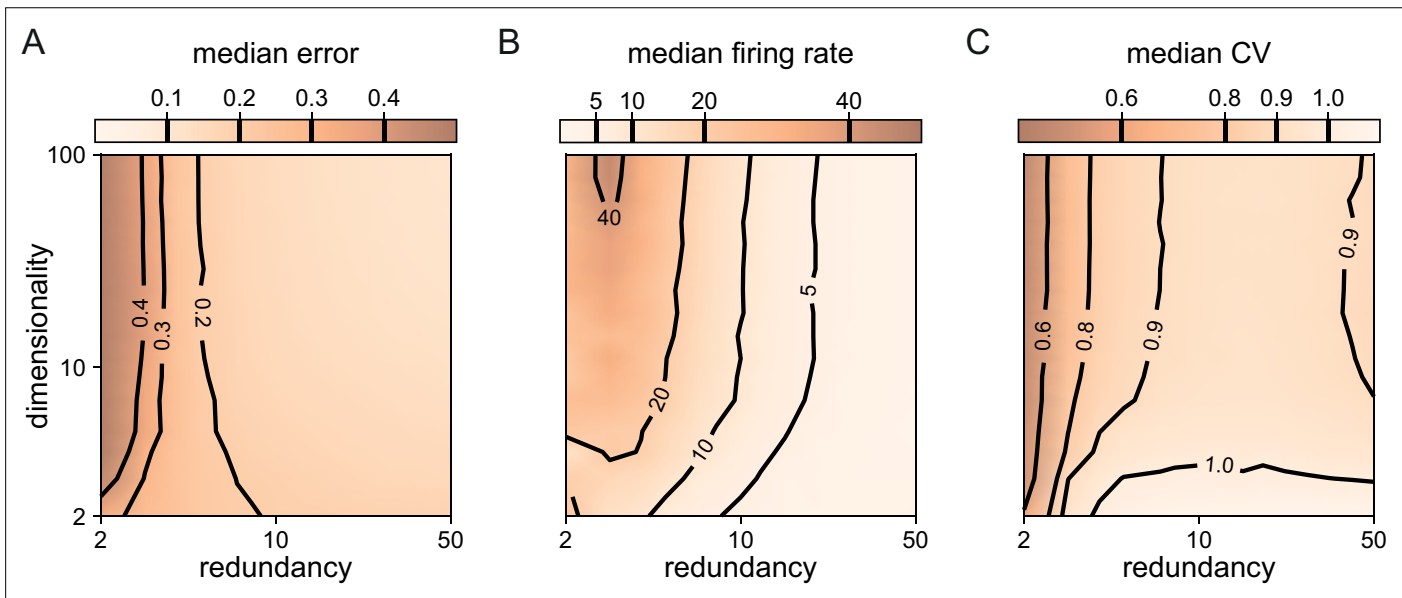

**Figure 5.** Median coding errors, firing rates, and CVs as a function of network redundancy and input dimensionality. All networks use random decoding vectors. (**A**) Most networks, except for very low redundancies, are able to correctly code for the signal. (**B**) Networks with low redundancy need to fire at higher rates, compared to networks with high redundancy, in order to keep the coding error in check. (**C**) Networks with low redundancy fire spikes in a more regular fashion (low CVs) compared to networks with high redundancy. Indeed, for networks with $\rho \approx 10$ and dimensionality $M \geq 3$, CVs are close to one, so that individual neurons produce spike trains with Poisson statistics.

to represent hundreds of signals simultaneously. To get closer to the biological reality, we therefore also need to study larger and more powerful networks. As we have shown above, the features of networks are tightly linked to the shape of the bounding box. For three-dimensional input signals, the threshold of each neuron becomes a plane, and the bounding box becomes a polyhedron (see also *Appendix 1—figure 3A* and *Video 1*). For higher dimensional signals, the precise shape of the bounding box is hard to visualize. However, if we assume that the number of neurons scales linearly with the number of signal dimensions, $N = \rho M$, one can show that the resulting, higher dimensional bounding boxes are somewhat closer to a hypercube than a hypersphere. (Some insights on the geometry of such higher dimensional bounding boxes can be found in *Appendix 1—figure 3*.)

In *Figure 3*, we saw that all bounding boxes are sensitive to excitatory perturbations, but that only no-redundancy (or very low-redundancy) bounding boxes are sensitive to inhibitory perturbations. A key question when scaling up is therefore whether larger networks with finite redundancy become sensitive to inhibitory perturbations, or whether they remain insensitive. An extreme form of inhibitory perturbation is the loss of neurons. *Figure 5* shows that, even in high dimensions, networks are robust against such perturbations. As before, we assume that the decoding vectors of the neurons $\mathbf{D}_i$ are of similar length, but otherwise random, and that the thresholds of all neurons are the same. As the death or birth of random neurons simply corresponds to a change in the overall redundancy of the network, we can understand how network performance and statistics change by simply moving along the redundancy axis in *Figure 5A–C*. We observe that changing the redundancy over a broad range ($\rho \approx 5 - 50$) has negligible effects on the performance (*Figure 5A*). This contrasts with networks of independent neurons in which performance scales linearly with any change in redundancy for a fixed readout. We furthermore observe that decreasing redundancy, leads to higher firing rates (*Figure 5B*) and more regular firing, or lower CVs (*Figure 5C*). The decrease of CVs here simply reflects a decrease in the number of spike patterns that can represent the constant input signal, given the smaller pool of neurons in the network. In other words, when we kill neurons, the neural code becomes less redundant, and the spike patterns of individual neurons lose some of their apparent randomness. Conversely, as the network size increases, so does the number of possible spike patterns, with the consequent increase of CV. As the number of neurons keeps increasing, it becomes more and more likely that the network has neurons optimally tuned to a given input signal, contributing to a decrease of the CV. Therefore, the increase and subsequent decrease in CV with increasing redundancy is the result of these two counteracting effects (*Figure 5C*).

In summary, when we scale up, networks with some redundancy remain robust to partial, inhibitory perturbations, even though the firing statistics of the neurons change.

## Natural perturbations

Biological systems should also be robust against the mistuning of any of their components. We will now show that many types of parameter mistuning can be understood as deformations of the bounding box. As shown in *Figure 3*, the simplest type of perturbation is a change in a neuron's spiking threshold: an increase of a neuron's spiking threshold will push the corresponding face of the bounding box outwards, and a decrease will push the face inwards.

While permanent changes in the threshold can come about through changes in conductances or reversal potentials, a neuron can also suffer from temporary changes in its effective spiking threshold through, for example, noise. Biological systems are constantly subject to noise at multiple levels such as sensory transduction noise, ion channel noise (*Faisal et al., 2008*), or 'background' synaptic activity (*Destexhe et al., 2001*; *Fellous et al., 2003*). We can study the impact of such noise by injecting small, random currents into each neuron. These currents change how close the voltage of a neuron is to its spiking threshold. With regard to spike generation, the resulting voltage fluctuations are thus equivalent to fluctuations of the threshold, or random movements of all of the faces of the bounding box around their unperturbed positions (*Figure 6A*, see also *Video 2*).

For networks with low redundancy, $\rho$, small voltage fluctuations cause only minor deformations of the bounding box. In turn, the error tolerance remains roughly the same, and network performance is not affected (*Figure 6B*, left; *Figure 6E and F*). However, for networks with high redundancy, $\rho$, small voltage fluctuations can cause a fatal collapse of the system (*Figure 6B*, middle). The key reason is that the effective size of the bounding box is determined by the position of the thresholds that have moved furthest into the box. As more and more neurons are added, the likelihood that some of them

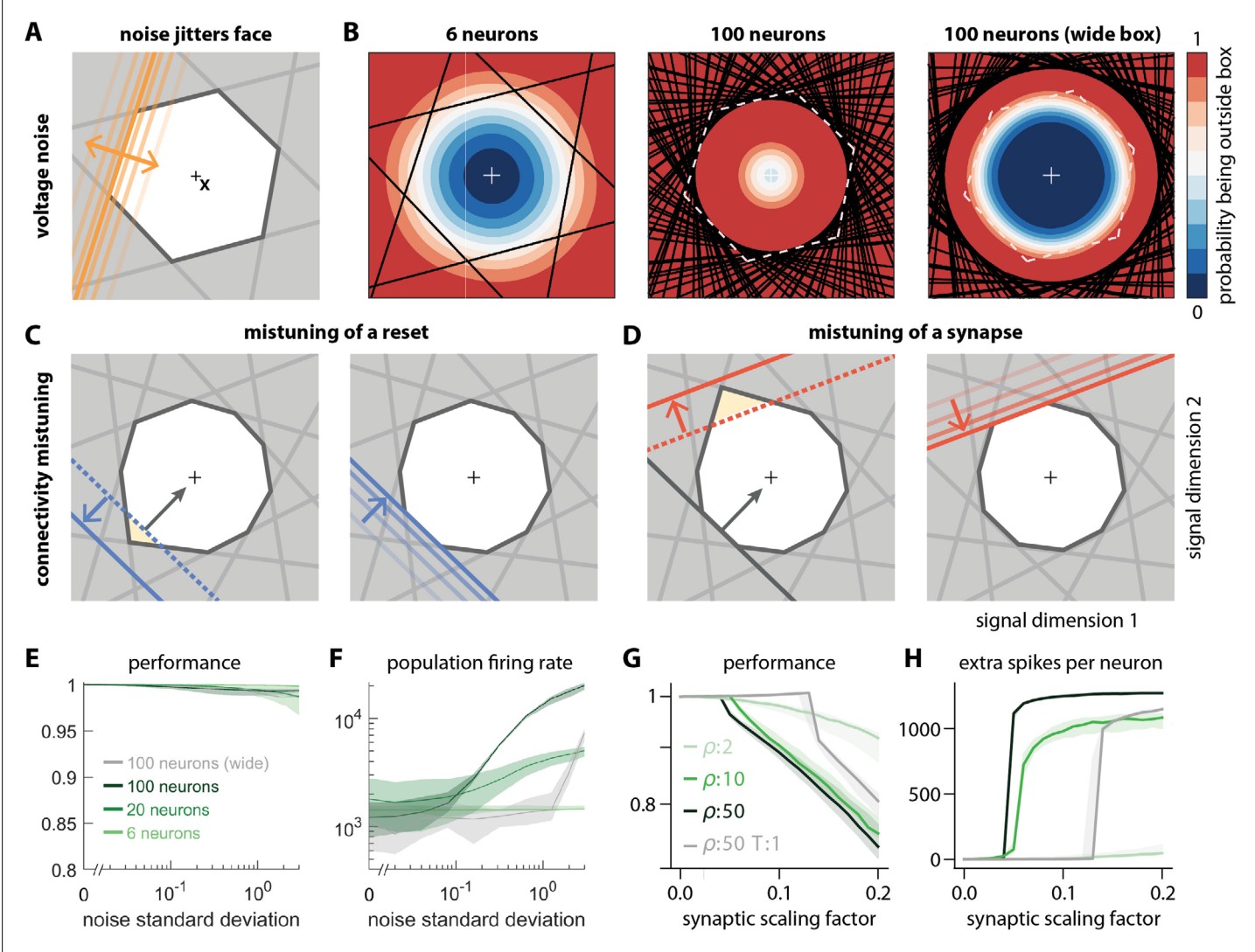

**Figure 6.** Network response to natural perturbations. (**A**) Voltage noise can be visualized as jittery movement of each threshold. If a neuron's threshold increases or decreases relative to its default value (solid orange), its respective boundary moves outward or inward. (**B**) Instead of a rigid box defining a permanent, unambiguous boundary between the spike and no-spike zones, any point in signal space now has a non-zero probability of falling outside the box, shown in color. Black lines represent the thresholds of individual neurons in the absence of noise. (left) At low redundancy, most points within the default box retain a low probability of exclusion. (centre) As redundancy increases, this low-probability volume disappears, increasing the likelihood of ping-pong spikes. (right) Networks with an expanded bounding box retain a large low-probability volume even at high redundancy. Dashed white lines show 6-neuron bounding box for comparison. (**C**) Temporary bounding box deformation caused by a mistuned reset. The deformation appears after a spike of the affected neuron and decays away with the time constant of the voltage leak. (**D**) Temporary bounding box deformation caused by a mistuned synapse. The deformation appears after a spike of the presynaptic neuron and decays away with the same time constant. (**E**) When noise level increases, performance (relative to an unperturbed network, see Methods, Network performance) drops only slightly. Lines show medians across random equidistant networks, and outlines represent interquartile ranges. (**F**) The ping-pong effect causes numerous unnecessary spikes for higher levels of noise, with more redundant networks affected more strongly. Networks with an expanded box retain healthy dynamics until much higher noise levels. (**G,H**) Each synapse is rescaled with a random factor taken from the interval $[1 - \delta_\Omega, (1 - \delta_\Omega)^{-1}]$, where $\delta_\Omega$ is the maximal synaptic scaling factor (see Materials and methods, Synaptic perturbations'). Networks are initially robust against synaptic mistuning, but eventually performance degrades. Networks with higher redundancy are more sensitive to these perturbations, but, as in the case of voltage noise, this extra sensitivity can be counteracted by widening the box.

have very decreased thresholds increases, and the effective size of the bounding box shrinks. In turn, the probability that the network moves into an 'epileptic seizure' (due to the 'ping-pong' effect, see *Appendix 1—figure 1*) increases as well. While the readouts may still be contained in this scenario (*Figure 6E*), the excessive number of spikes fired (*Figure 6F*) comes at a high metabolic cost and

would be detrimental to biological systems. To avoid this failure mode, neurons need to lower their excitability, which in turn increases the size of the bounding box for a fixed redundancy (*Figure 6B*, right panel). Such a 'wide box' will be more resilient towards noise (*Figure 6B*, right panel, *Figure 6E and F*). More generally, our results suggest that more redundant networks may require a better control or suppression of intrinsic sources of noise than less redundant networks.

Next, we will study perturbations of a neuron's reset potential, that is, the voltage reached directly after a spike. This voltage should ideally be $V_{i,\mathrm{reset}} = T_i - \mathbf{D}_i^\top \mathbf{D}_i$. Biophysically, when the neuron resets to a voltage above (below) this ideal reset potential, then its post-spike voltage is temporarily closer (further) from threshold. In terms of the neuron's spiking output, a change in its reset voltage is therefore equivalent to a (temporary) change in its threshold. Within the bounding box, a reset voltage above (below) the optimal reset will lead to a push of the neuron's threshold inwards (outwards) (*Figure 6C*). However, because of the voltage leak, the threshold will then decay back to its normal position. *Video 2* illustrates this effect in a system with a two-dimensional input. We note that positive and negative changes to the default reset potential will lead to asymmetric effects on robustness like those observed for excitatory and inhibitory perturbations. Specifically, if the resets become too small, and if the leak is insufficiently fast, then successive spiking of a single neuron will draw its threshold inwards, thereby leading to a collapse of the bounding box.

Finally, we study perturbations of the synaptic connectivity in the network. Synapses could be permanently mistuned or they could be temporarily mistuned, for instance through transmission failures or through stochastic fluctuations in the release of neurotransmitters (*Faisal et al., 2008*). From a geometric perspective, a mistuned synapse causes a temporary change in the threshold of the postsynaptic neuron whenever a presynaptic spike arrives (*Figure 6D*). We again note an asymmetry: an excitatory synapse with decreased strength (or an inhibitory synapse with increased strength) leads to an outward move of the postsynaptic neuron's threshold, which is generally harmless. In turn, an excitatory synapse with increased strength (or an inhibitory synapse with decreased strength) leads to an inward move, which could be a temporarily harmful perturbation. Accordingly, random synaptic failures in excitatory synapses (but not inhibitory synapses) leave the bounding box functionally intact.

When all synapses in the network are randomly mistuned, then each spike fired will cause a random, but transient deformation of the bounding box (see *Video 2*). Overall, we find that more redundant networks (with consequently more synapses) are typically more vulnerable to these perturbations. Just as for voltage noise, the amount of deformation of the bounding box therefore increases with the number of neurons. For large perturbations, the synaptic noise eventually leads to inefficient networks with high spike rate (*Figure 6G and H*). As shown in *Appendix 1—figure 4*, the effects of (voltage or synaptic) noise on the networks hold independent of the signal dimensionality.

## Synaptic delays

So far, we have assumed that the propagation of action potentials is instantaneous. However, lateral excitation and inhibition in biological networks incur delays on the order of milliseconds. Previous work has shown that networks which coordinate their spiking as suggested here are extremely sensitive to delays when neurons are similarly tuned (*Chalk et al., 2016*; *Rullán Buxó and Pillow, 2020*). Indeed, when spikes are delayed, voltages no longer reflect an accurate estimate of the coding error. For neurons with identical decoders, delays can lead to uninformed spikes that actually increase the coding error (*Figure 7A and B*). With networks that represent M-dimensional signals at once, the effects of delays are more complex. However, the bounding box allows us to visualize them and explain how they can, in principle, be avoided. Below, we study the impact of these delays, which apply directly to recurrent excitation and inhibition. We also apply the same delays to the network readout for mathematical convenience, but those do not affect the network dynamics (see Materials and methods).

To visualize the effect of a synaptic delay, we show the readout dynamics around a single spike in *Figure 7C* (see also *Video 2*). After hitting the threshold, the spiking neuron resets its own voltage immediately. However, due to the delay, neither a hypothetical readout unit nor other neurons in the network are aware of the spike. From the network perspective, the voltage of the spiking neuron appears temporarily too low (or its threshold too high), which we can visualize as an outward jump of its boundary (*Figure 7C*, second and third panels). When the spike finally arrives, the readout and all

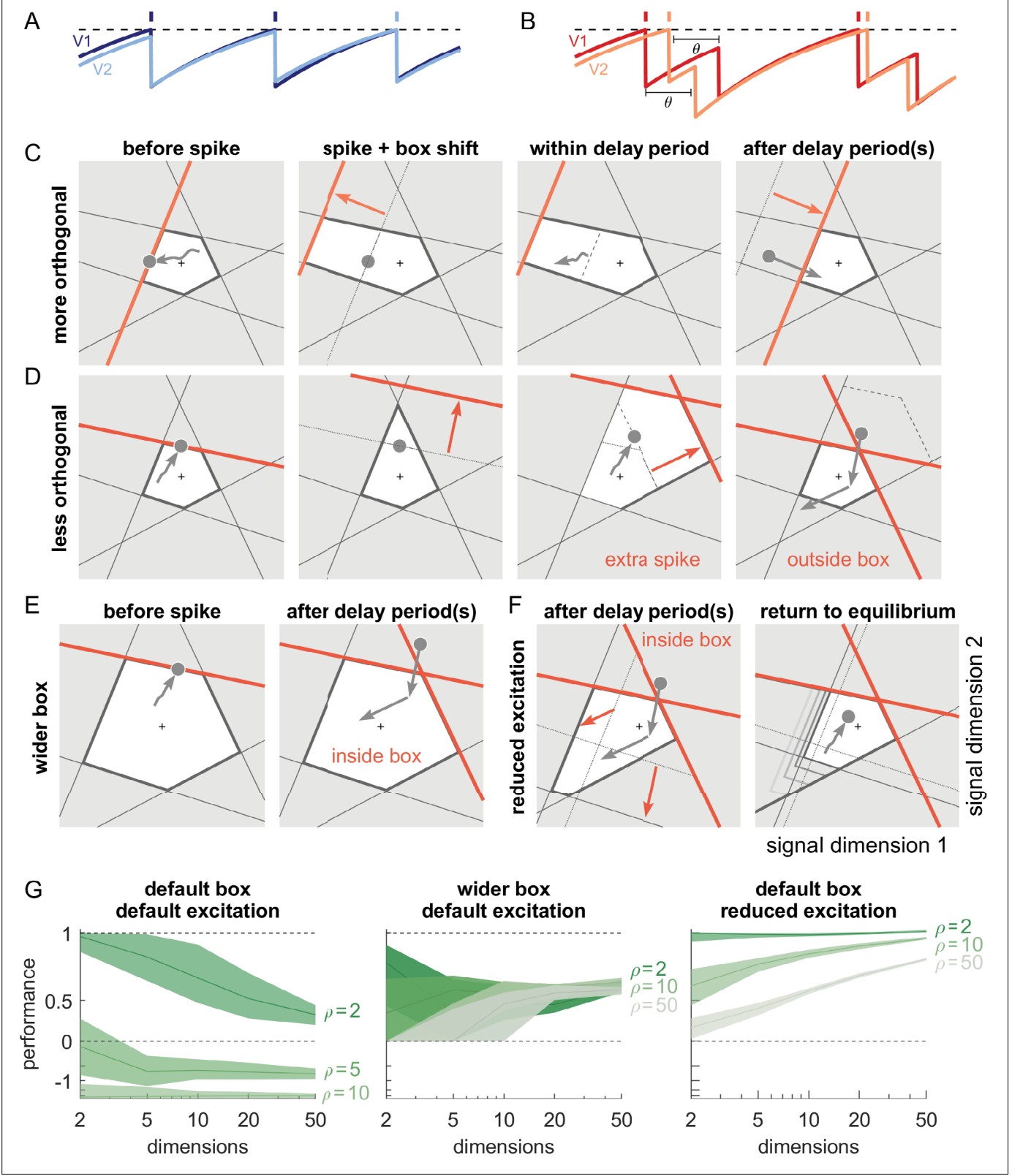

**Figure 7.** Synaptic transmission delays cause uninformed spikes, but networks with high-dimensional inputs are less affected. (**A**) In an undelayed network, when membrane potentials $V_1$ and $V_2$ of two identically tuned neurons approach firing threshold (dashed), the first neuron to cross it will spike and instantly inhibit the second. (**B**) If recurrent spikes are instead withheld for a delay $\theta$, the second neuron may reach its own threshold before receiving this inhibition, emitting an 'uninformed' spike. (**C**) Readout dynamics in a delayed network that encodes a two-dimensional input. After the

*Figure 7 continued on next page*

Figure 7 continued

spike of the orange neuron, but before its arrival at synaptic terminals, the voltage of the orange neuron is temporarily too low, causing an effective retraction of its boundary. (**D**) For less orthogonal pairs of neurons, the retraction of the boundary of a spiking neuron may expose the boundary of a similarly tuned neuron, leading to a suboptimally fired spike, and increasing the likelihood of 'ping-pong'. (**E**) Permanently wider boxes or (**F**) temporarily wider boxes (excitatory connections between opposing neurons removed) are two effective strategies of avoiding 'ping-pong'. (**C–F**) Readout shown as gray circles and arrows, bounds of spiking neurons as colored lines, and the resulting shift of other bounds as colored arrows. (**G**) Simulations of networks with a synaptic delay of $\theta = 1$ msec. (Left) In standard networks, performance quickly degenerates when redundancy is increased. (Centre, Right) The detrimental effects of delays are eliminated in higher-dimensional bounding boxes that are widened (centre) or when the largest excitatory connections are removed (right). Note the exponential scaling of the y-axis. See *Appendix 1—figure 5* for single trials with normal or wide boxes, and full or reduced connectivity (20 dimensions).

other voltages are updated, and the voltage of the firing neuron once again agrees with the network state. In our visualization, its boundary thus returns to its default position (*Figure 7C*, fourth panel).

The visualization illustrates that the effect of a delayed spike depends on the bounding box shape. In *Figure 7C*, the nearly orthogonal tuning of neighbouring neurons makes the delayed spike harmless. The situation is different if neurons are more similarly tuned as in *Figure 7D*. Here, a second neuron might cross its threshold before the delayed spike arrives. As a consequence, it also fires a spike, and its boundary also retracts (*Figure 7D*, third panel). Eventually, both spikes arrive, the readout is updated, and the bounding box regains its original shape (*Figure 7D*, fourth panel). At this point, the readout may overshoot, cross an opposite boundary, and trigger further 'ping-pong' spikes. The resulting 'epileptic seizures' are essentially unavoidable in highly redundant networks with synaptic delays. Consequently, we identify two problems with synaptic delays. The first problem is that multiple thresholds are crossed simultaneously. The second problem is that the resulting strong change in the readout can cause a 'ping-pong' of uninformed spikes.

For the first problem, we note that the impact of synaptic delays depends on the angles of neighbouring neurons, as shown in *Figure 7C and D*. For higher signal dimensions and fixed redundancy, these angles become more orthogonal (*Appendix 1—figure 3D*), which alleviates the detrimental effect of delays. Numerically, however, we find that this effect is not sufficient to avoid all uninformed spikes (for signal spaces with up to $M = 50$ dimensions), and the networks still degenerate into 'ping-pong'.

To avoid the second problem, we need to eliminate the crossing of opposite thresholds by widening the box, which can prevent 'ping-pong' (*Figure 7E*, *Appendix 1—figure 1*). However, permanently widening the bounding box in all directions can reduce coding accuracy, even when the readout is properly rescaled (*Figure 7G*, *Appendix 1—figure 5*, see Materials and methods, 'Iterative adaptation of parameters to avoid ping-pong'). A better solution is therefore to 'widen' the box only temporarily. For instance, if we eliminate excitatory connections between pairs of neurons that are practically antipodes, we are setting the respective synapses to zero. This change in the network connectivity can be understood as a specific and targeted synaptic perturbation (*Figure 6D*), whose effect is to expand the thresholds of a neuron's antipodes whenever it fires a spike, thereby temporarily widening the box exactly in the direction in which the readout overshoots (*Figure 7F*). As a consequence, the networks become less likely to initiate ping-pong. Moreover, as their widening is local and only temporary, performance is less affected. Indeed, for higher dimensional systems and biologically plausible delays (1–2 ms), performance of networks with delays reaches the performance of networks without delays (*Appendix 1—figure 5*). The rapid increase in firing due to ping-pong is avoided as well (see also *Appendix 1—figure 5*).

## Adding computations

The bounding box provides a useful tool even if we endow the networks with a set of slower connections to perform linear or non-linear computations (*Boerlin et al., 2013*; *Savin and Deneve, 2014*; *Thalmeier et al., 2016*). Indeed, the simulation in *Figure 1D* used these slower connections to generate oscillatory dynamics (see Materials and methods, 'Generalization of the bounding box IV'). This extension to networks that generate persistent activity or dynamical patterns works because the mechanisms underlying the encoding of the signals into spike trains are decoupled from the mechanisms that generate the dynamics of the signals (or readouts). In fact, the extra currents generated by

the slow recurrent connections can be seen as a perturbation of the bounding box thresholds. This perturbation shifts the bounding box in the space of readouts as illustrated in *Appendix 1—figure 6*.

## Discussion

In this study, we characterized the functioning of networks with coordinated redundancy under normal conditions and under a diversity of perturbations, using a simple, geometric visualization, the bounding box. The bounding box delimits the error that such a network tolerates in approximating a set of input signals, and its geometry is found to be largely determined by the properties of its decoders. It allows us to visualize and thus understand the dynamics of coordinated spiking networks, including the firing of every single spike. We showed how various perturbations of the network can be mapped onto shape deformations of this bounding box. As long as the box stays intact, the network's performance is essentially unaffected, in that downstream readouts of the network's outputs will not notice the perturbation.

In many respects, the bounding box is a 'toy model' (a deliberately abstract model), which we see mainly as a tool to conceptualize and highlight generic circuit mechanisms, rather than an attempt to model any specific system. Nonetheless, it is worthwhile to point out that the bounding box is also a spiking version of classical sparse coding models of V1 (*Olshausen and Field, 1996*). Indeed, previous work has demonstrated that these networks can explain various perturbation experiments in V1 (*Barrett et al., 2016*). So, besides shedding light on the robustness of coordinated spike codes, the bounding box can also be seen as a simple model of a sensory system.

### Robustness of networks with coordinated spike coding

Several overarching principles have been identified that allow systems to be robust (*Csete and Doyle, 2002*; *Kitano, 2004*; *Whitacre, 2012*; *Félix and Barkoulas, 2015*). These include (1) negative feedback, to correct perturbations and recover functionality; (2) heterogeneity of components, to avoid common modes of failure; and (3) modularity or 'bow-tie' architectures, to create alternative pathways or solutions in the case of a perturbation. Furthermore, (4) making a system robust against certain perturbations almost always involves a tradeoff, in that the system becomes fragile against other perturbations.

These core themes can also be found in the networks we studied here. (1) Negative feedback exists through extensive lateral connectivity (or, alternatively, through actual feedback of the readout, as in *Figure 2F*), and is precisely tuned such that it automatically corrects any perturbations. (2) Individual neurons are heterogeneous and thereby allow the system (as visualized by the bounding box) to remain functional for all types of input signals. (3) Since neuron space is always larger than signal space, there are many alternative neural codes ('alternative pathways') that give rise to the same linear readout, thus embodying a bow-tie architecture whose core is the signaling space. Shrinking the network's redundancy, for example, by killing neurons, in turn eliminates these alternative codes and leads to more regular and reliable spike trains. (4) Furthermore, the networks are fragile against any perturbation that leads to a shrinking of the box. Paradoxically, this fragility may become more relevant if a system becomes more redundant. These four themes may relate the robustness of the networks studied here to the more general topic of tissue robustness (*Kitano, 2004*).

Coordinated redundancy allows the construction of robust sub-circuits, that can self-correct problems instead of passing them on, so that downstream networks remain unaffected. These observations remain correct even if we move beyond the simple autoencoder networks that we have studied here. Indeed, we could generalize the connectivities we consider or abandon the idea that the readout must match the input without changing the robustness of the networks (see Materials and methods, 'Coordinated spiking and the bounding box'). We could also add slower recurrent synapses which allows to generate dynamics within the networks (*Boerlin et al., 2013*; *Savin and Deneve, 2014*; *Thalmeier et al., 2016*), as explained above.

### Fragility of networks with coordinated spike coding

Despite their strong robustness, networks with coordinated redundancy are also surprisingly fragile against any perturbations that cause an effective shrinking of the box, and thereby lead to the ping-pong effect. These problems can be ameliorated by widening the box, which brings networks back

into workable regimes if they represent high-dimensional signals with limited redundancy. However, a true 'fix' of this problem can only be achieved if neurons with opposite decoder weights (which are connected through excitatory connections) are prohibited. Such a change would break the symmetric treatment of excitatory and inhibitory connections, which causes neurons to both excite and inhibit different downstream partners, thereby violating Dale's law. Future work will need to reconsider these issues which seem to be tightly connected. (We note that *Boerlin et al., 2013* developed networks that obey Dale's law, but did so without fixing the issue of the ping-pong effect.)

## Structural robustness of neural networks

Historically, the study of network-level mechanisms of robustness has received relatively little attention. A key focus has been the robustness of network attractors, defined as the ability of a system to remain in the same attractor landscape despite perturbations. For instance, systems such as the oculomotor integrator or the head direction system can be described as continuous attractors (*Seung, 1996*; *Zhang, 1996*). Such continuous attractors are structurally unstable, in that even small perturbations in single neurons can lead to rapid dynamic drifts (*Seung, 1996*; *Zhang, 1996*). However, this fragility to perturbations is not observed in biological neural networks.

In order to achieve the required robustness, several biophysical mechanisms have been proposed to enhance continuous attractors models, e.g. bistability at the somatic level (*Koulakov et al., 2002*) or dendritic level (*Goldman et al., 2003*). More recent work proposed network-level mechanisms based on derivative feedback, in order to solve the problem of robustness for continuous attractor networks (*Lim and Goldman, 2013*). In our work, the problem is solved because perturbations such as neuron loss, noise, or tuning of synapses are compensated through the fast, lateral connections. As a consequence, perturbations of the single-neuron level (spiking) are uncoupled from perturbations of the population-level (readout). Consequently, only perturbations that manage to disturb the linear readout can impact the network attractor dynamics.

Models of neural networks implementing point attractors, such as the Hopfield model (*Hopfield, 1982*), are typically considered structurally robust, meaning that perturbations up to certain magnitudes of their parameters and the introduction of dynamics noise do not disrupt the attractor. We note, however, that perturbations in these networks lead to changes in neurons' firing rates, which may still cause changes in putative downstream linear readouts. From the point of view of a downstream observer, perturbations are therefore not compensated within classical attractor networks. The induced perturbations may be inconsequential, however, when the downstream readout is taken to be a classifier; only the combined system of attractor network and classifier readout can then be seen as a 'robust module', that is, a module that keeps problems to itself, rather than spreading them to all who listen.

Similar observations apply to studies of the robustness of deep networks against various perturbations such as the loss of neurons (*Morcos, 2018*; *Barrett et al., 2019*). In these cases, the network's robustness is evaluated with respect to the output of a final classification step, such as the identification of an object. Indeed, a lot of work has been dedicated to making this final output robust to small perturbations, especially perturbations applied to the inputs (*Szegedy, 2013*; *Biggio, 2013*; *Carlini, 2019*; *Brendel et al., 2020*). Based on the arguments above, we similarly expect that the problem of making a graded output robust will be harder and fundamentally different.

## Insights on spiking networks

Spiking networks have traditionally been quite hard to understand, except for special cases (*Maass and Bishop, 1999*; *Vogels et al., 2005*; *Gerstner et al., 2014*). Here, we have shown how the dynamics of (coordinated) spike coding networks can be understood within a lower-dimensional signal space, which is tightly linked to linear readouts. Since (low-dimensional) linear readouts are a ubiquitous finding in recordings from neural populations, we may speculate that our signal space is roughly equivalent to the latent subspaces discovered by linear projections of neural activities, as, for example, obtained through dimensionality reduction methods (*Cunningham and Yu, 2014*; *Keemink and Machens, 2019*). This link between a space of neural activities and a space of (latent) signals is common to all network models based on low-rank connectivities (*Eliasmith, 2005*; *Seung, 1996*; *Mastrogiuseppe and Ostojic, 2018*). In contrast to these studies, however, and in line with (*Boerlin et al., 2013*), our work focuses on spiking networks and introduces a third space, the voltage space,

which represents the system's coding errors. As we have shown here, the coding errors are confined to an error bounding box. Accordingly, the bounding box finds its physical—and in principle measurable—manifestation in a low-dimensional subspace of a network's voltage space (*Appendix 1—figure 2*).

We believe that the links we have made here—which allow us to jointly visualize a low-dimensional signal space, the spiking activity, and the subthreshold voltages—may provide useful insights into the functioning of spiking networks in the brain, and may well be expanded beyond the confines of the current study.

## Materials and methods
### Coordinated spiking and the bounding box

Mathematically, our networks can be derived from a single objective function that quantifies coding accuracy. Step-by-step derivation for the autoencoder networks can be found in *Barrett et al., 2016*; networks that additionally involve a set of slow connections are derived in *Boerlin et al., 2013*. Here, we focus on the autoencoder networks which contain all the crucial elements needed to understand the spiking dynamics of the networks. Instead of starting with an objective function, we take a slightly different perspective in our derivation here, which ties more directly into our geometric interpretations.

In short, we assume that a network of $N$ neurons encodes an $M$-dimensional input signal $\mathbf{x}(t)$, in its spike trains $\mathbf{s}(t)$, such that the signal can be read out from the filtered spike trains,

$$\hat{\mathbf{x}}(t) = \mathbf{D}\mathbf{r}(t) \tag{4}$$

$$\dot{\mathbf{r}}(t) = -\lambda\mathbf{r}(t) + \mathbf{s}(t). \tag{5}$$

Here, $\hat{\mathbf{x}}(t)$ is the $M$-dimensional linear readout or signal estimate, the $M \times N$ matrix $\mathbf{D}$ contains the decoding weights (and each column corresponds to a decoding vector $\mathbf{D}_i$), the filtered spike trains are represented by an $N$-dimensional vector $\mathbf{r}(t)$, and $\lambda$ determines the filtering time constant.

The key idea of coordinated spike coding is to derive a spiking rule that bounds the difference between the input signal $\mathbf{x}$, and the linear readout $\hat{\mathbf{x}}$,

$$\|\mathbf{x} - \hat{\mathbf{x}}\| < T, \tag{6}$$

where $\|\cdot\|$ denotes the Euclidean distance or L2 norm and $T$ determines the maximally allowed difference. In the network implementation, we approximate this bound (which defines a hypersphere) by a set of linear bounds or inequalities, one for each neuron $i$,

$$\mathbf{D}_i^\mathsf{T}(\mathbf{x} - \hat{\mathbf{x}}) < T. \tag{7}$$

For simplicity, we assume that the decoding vectors $\mathbf{D}_i$ have unit norm. Each inequality defines a half-space of solutions for the readout $\hat{\mathbf{x}}$. For properly chosen $\mathbf{D}_i$, the intersection of all of these half-spaces is non-empty and bounded, and thus forms the interior of the bounding box. Geometrically, the equations define a polytope $B = \{\hat{\mathbf{x}} \in \mathbb{R}^M \mid \mathbf{D}^\mathsf{T}(\mathbf{x} - \hat{\mathbf{x}}) < \mathbf{T}\}$. If the thresholds are chosen sufficiently large, then crossing a bound and firing a spike keeps the readout inside the bounding box.

The dynamics of the network are obtained by identifying the left-hand side of the above equation with the neuron's voltage, $V_i$, and then taking the temporal derivative (*Boerlin et al., 2013*; *Barrett et al., 2016*). If we also add some noise to the resulting equations, we obtain,

$$\dot{\mathbf{V}} = -\lambda\mathbf{V} + \mathbf{D}^\mathsf{T}\left(\lambda\mathbf{x}(t) + \dot{\mathbf{x}}(t)\right) - \mathbf{D}^\mathsf{T}\mathbf{D}\mathbf{s}(t) + \sigma_V\boldsymbol{\eta}(t), \tag{8}$$

which describes a network of leaky integrate-and-fire neurons. The first term on the right-hand side is the leak, the second term corresponds to the feedforward input signals to the network, the third term captures the fast recurrent connectivity, with synaptic weights $\Omega_{ij} = -\mathbf{D}_i^\mathsf{T}\mathbf{D}_j$, and the fourth term is added white current noise with standard deviation $\sigma_V$. When the voltage $V_i$ reaches the threshold $T$, the self-connection $\Omega_{ii} = -\mathbf{D}_i^\mathsf{T}\mathbf{D}_i$ causes a reset of the voltage to $V_{\text{reset}} = T + \Omega_{ii}$. For biological plausibility, we also consider a small refractory period of $\tau_{\text{ref}} = 2\text{ms}$ for each neuron. We implemented this refractory period by simply omitting any spikes coming from the saturated neuron during this period.

Mathematically, the voltages are thereby confined to a subspace given by the image of the transposed decoder matrix, $\mathbf{D}^\top$. The dynamics within this voltage subspace are then bounded according to

*Equation 7*, which can be seen as a physical manifestation of the bounding box (see also *Appendix 1—figure 2*).

## Generalization of the bounding box I: Heterogeneous thresholds

In our exposition, we generally assume that all decoding vectors are of the same length, and all thresholds are identical. For isotropically distributed input signals and isotropically distributed decoding vectors, this scenario will cause all neurons to fire the same average number of spikes over time. Indeed, to the extent that homeostatic plasticity sets synaptic weights and firing thresholds to guarantee this outcome (*Turrigiano, 2012*), a network will automatically revert to a spherically symmetric bounding box for such input signals (see also *Brendel et al., 2020*).

If input signals are not isotropically distributed then homeostatic plasticity would essentially lower the thresholds of neurons that receive overall less inputs, and it would increase the thresholds of neurons that receive overall more inputs. In turn, the bounding box would take more elliptical shapes. We have not considered this scenario here for simplicity, but the key findings on robustness will hold in this case, as well.

## Generalization of the bounding box II: Asymmetric connectivities

In the main text, we have assumed that the readout always jumps orthogonal to the threshold boundary (or face) of a neuron. This assumption leads to symmetric connectivities in the network, given by $\Omega_{ij} = -\mathbf{D}_i^\mathsf{T} \mathbf{D}_j$. However, our results on robustness also hold if we decouple the orientation of a neuron's face from the direction of the readout jump. This can be achieved if we define the voltage as $V_i = \mathbf{F}_i(\mathbf{x} - \hat{\mathbf{x}})$, where $\mathbf{F}_i$ denotes the norm vector of a bounding box face, but then let the readout jump in the direction $\mathbf{D}_i$. A non-orthogonal jump with respect to the face then simply requires $\mathbf{D}_i \neq \mathbf{F}_i$. Indeed, for elliptically shaped bounding boxes, non-orthogonal jumps of the readout can be advantageous. The more general dynamical equation for the networks is then given by

$$\dot{\mathbf{V}} = -\lambda \mathbf{V} + \mathbf{F}\left(\lambda \mathbf{x}(t) + \dot{\mathbf{x}}(t)\right) - \mathbf{F}\mathbf{D}\mathbf{s}(t) + \sigma_V \boldsymbol{\eta}(t), \tag{9}$$

and was first described in *Brendel et al., 2020*. In principle, these generalized networks include all spiking networks with low-rank connectivities. However, the bounding box interpretation is most useful when each spike is reset back into the bounding box, which will only happen if the net effect of a spike on neighboring neurons is inhibitory. Spikes that cause (temporary) jumps out of the box, and therefore have a net excitatory and error-amplifying effect, will be considered in future work.

## Generalization of the bounding box III: Opening the box

The equation for the synaptic connectivity, $\Omega_{ij} = -\mathbf{D}_i^\mathsf{T} \mathbf{D}_j$, implies that neurons with similar decoding vectors inhibit each other, neurons with orthogonal decoding vectors are unconnected, and neurons with opposite decoding vectors excite each other. Consequently, if the bounding box is a (hyper) cube, then almost all neurons are unconnected, except for neurons whose faces are opposite to each other. The excitatory connections between these neurons ensure that their voltages remain in sync. However, in practice, those voltages do not need to be tied, and the excitatory connections can therefore also be eliminated (as in *Figure 7*), which can help against the ping-pong effect.

Alternatively, we can choose decoding vectors such that all synapses are inhibitory, $\Omega_{ij} \leq 0$. In this case, the bounding box remains open on one side. The network no longer represents the input signal, but rather computes a piece-wise linear function of the input (*Mancoo, 2020*). In turn, the network's new function (piece-wise linear output) will now remain robust against perturbations for exactly the same reasons explained before. Indeed, the reader may notice that most of the results on robustness do not require the bounding box to be closed.

## Generalization of the bounding box IV: Slow connections

Throughout the manuscript, we focused on autoencoder networks. However, as illustrated in *Figure 1* and derived in *Boerlin et al., 2013*, by introducing a second set of slower connections, we can endue these networks with computations,

$$\dot{\mathbf{V}} = -\lambda \mathbf{V} + \mathbf{D}^\mathsf{T}\left(\lambda \mathbf{x}(t) + \dot{\mathbf{x}}(t)\right) - \mathbf{D}^\mathsf{T}\mathbf{D}\mathbf{s}(t) + \mathbf{D}^\mathsf{T}\left(\mathbf{A} + \lambda\mathbf{I}\right)\mathbf{D}\mathbf{r}(t) + \sigma_V \boldsymbol{\eta}(t), \tag{10}$$

in which case the network approximates the dynamical system:

$$\dot{\mathbf{x}} = \mathbf{A}\mathbf{x} + \mathbf{c}(t).$$

We note that all results on robustness hold for these more complicated networks as well. Indeed, the robustness of the autoencoder networks relies on the fast recurrent connections, which are present in these architectures as well. Due to the time scale separation, these mechanisms do not interfere with the slower recurrent connections, which create the slow dynamics of the readouts (see also *Appendix 1—figure 6*).

## Readout biases and corrections

When one of the neurons fires, its spike changes the readout, which jumps into the bounding box. In previous work (*Boerlin et al., 2013*, *Barrett et al., 2016*), the neurons' thresholds were linked with the length of the jumps through the equation $T_i = \|\mathbf{D}_i\|^2/2$. Accordingly, the jumps were generally taken to reach the opposing face of the bounding box, creating a tight error bounding box around $\mathbf{x}$. This setting guarantees that the time-averaged readout matches the input signal.

When the jumps are significantly shorter than the average bounding box width, however, the time-averaged readout will be biased away from the input signal (see *Appendix 1—figure 1*). However, in many cases, this bias can be corrected by rescaling the readout. For instance, if the bounding box is shaped like a hypersphere (i.e. in the limit of an infinite number of neurons $N$), and assuming a constant (or slowly varying) stimulus, we can correct the readout as

$$\hat{\mathbf{x}} = \left( \frac{\langle\|\mathbf{D}\mathbf{r}\|\rangle + T - \frac{1}{2}}{\langle\|\mathbf{D}\mathbf{r}\|\rangle} \right) \mathbf{D}\mathbf{r}, \tag{11}$$

where the angular brackets denote the time-averaged readout. Accordingly, in this case the bias only affects the length of the readout vectors, but not their direction.

If the bounding box is shaped like a hypercube, we alternatively correct the readout bias by assuming that a downstream decoder area has access to the identity of spiking neurons in the recent past. In this case, the downstream area can simply correct the readout according to the following equation:

$$\hat{\mathbf{x}} = \mathbf{D}\mathbf{r} + \sum_{i \in S} \left( \frac{1}{2} - \frac{T}{\|\mathbf{D}_i\|^2} \right) \mathbf{D}_i, \tag{12}$$

where $S$ is the set of active neurons for a given fixed time window in the past.

In all other cases, we empirically found that we can apply a correction to the readout using a similar scaling as in *Equation 11* where $\langle\|\mathbf{D}\mathbf{r}\|\rangle \approx \|\mathbf{D}\mathbf{r}(t)\|$. In other words, in most cases, the bias mainly affects the length of the readout vectors, whereas their direction is less affected.

In *Figure 1*, we used networks that involve an extra set of slow recurrent connections (*Boerlin et al., 2013*). In this case, we additionally scaled the slow recurrent connectivity matrix $\Omega_{\text{slow}}$ with the same scaling factor as the corrected readout in *Equation 11*:

$$\Omega_{\text{slow}} = \left( \frac{\langle\|\mathbf{D}\mathbf{r}\|\rangle + T - \frac{1}{2}}{\langle\|\mathbf{D}\mathbf{r}\|\rangle} \right) \mathbf{D}^{\mathsf{T}} \left( \mathbf{A} + \lambda\mathbf{I} \right) \mathbf{D}. \tag{13}$$

## Geometry of high-dimensional bounding boxes

The dimensionality of the bounding box is determined by the dimensionality $M$ of the input signal. Throughout the illustrations in the Results section, we mostly used two-dimensional bounding boxes for graphical convenience. In order to illustrate some properties of higher-dimensional error bounding boxes (*Appendix 1—figure 3*), we compared their behavior against that of hyperspheres and hypercubes. We defined the equivalent hypersphere as $\{\boldsymbol{p} \in \mathbb{R}^M : \|\boldsymbol{p}\|_2 \leq T\}$ and the equivalent hypercube as $\{\boldsymbol{p} \in \mathbb{R}^M : \|\boldsymbol{p}\|_\infty \leq T\}$, where $\|\boldsymbol{p}\|_2 = \sqrt{p_1^2 + \ldots + p_n^2}$ and $\|\boldsymbol{p}\|_\infty = \max_i |p_i|$. In practice, we chose the smallest box size, $T = 0.5$ (*Appendix 1—figure 3*).

For a first comparison, we took the intersection between the border of the $M$-dimensional polytope $B$ and a random two-dimensional plane containing the centre of the polytope. We computed such intersections numerically by first choosing two random and orthogonal directions $u$ and $v$ in the

full space defining the two-dimensional plane. Then for each $\theta \in [0, 2\pi]$, we defined a ray in the two-dimensional plane, $w(\rho) = \rho\cos(\theta)u + \rho\sin(\theta)v$, and then plotted

$$\rho(\theta) = \underset{\rho > 0, w(\rho) \in B}{\arg\max} w(\rho).$$

For a second comparison, we found the distribution of angles between neighbouring neurons by first randomly choosing one neuron, and then moving along the surface of the $M$-polytope in a random direction, until we found a point that belongs to the face of a different neuron. We then computed the angle between the decoding weights of those two neurons.

We tested whether the results obtained for random decoding vectors hold for more structured decoding vectors as well. For instance, if we want to represent natural visual scenes, we may consider that the receptive fields of simple cells in V1 roughly correspond to the decoding vectors of our neurons (*Olshausen and Field, 1996*; *Barrett et al., 2016*). We illustrated a high-dimensional bounding box with a set of Gabor patches defined as

$$g(x, y; \lambda, \theta, \sigma, \gamma) = \exp\left(-\frac{\tilde{x}^2 + \gamma^2 \tilde{y}^2}{2\sigma^2}\right) \cos\left(2\pi \frac{\tilde{x}}{\lambda} + \frac{\pi}{2}\right), \tag{14}$$

where $\tilde{x} = x\cos\theta + y\sin\theta$ and $\tilde{y} = -x\sin\theta + y\cos\theta$. For our purposes, we randomly chose the Gabor parameters: $\lambda$, the wavelength of the sinusoidal stripe pattern, was sampled uniformly from $\{3, 5, 10\}$ Hz; $\theta$, the orientation of the stripes, was sampled uniformly in $[0, 2\pi]$, the standard deviation of the Gaussian envelope, was sampled uniformly from $\{1, 1.5\}$, the spatial aspect ratio, was sampled uniformly from $\{1, 1.5\}$.

Finally we randomly centred the resulting Gabor patch in one of 9 different locations on the $13 \times 13$ grid. We computed the angle (in the 169-dimensional space) between the Gabor patches and found that roughly 80% of the neurons are quasi-orthogonal (their angle falls between 85 and 95 degrees) to a given example patch (*Appendix 1—figure 3E*).

## Perturbations

Perturbations to the excitability of a neuron, be it due to changes of the spiking threshold, changes of the reset potential, synaptic weights, etc., can all be formulated as extra currents, $p_i(t)$, which capture the temporal evolution of the perturbation. Adding a current to the voltage dynamics is equivalent to a transient change in the neuronal thresholds,

$$\begin{aligned} \dot{\mathbf{V}} &= -\lambda\mathbf{V} + \mathbf{D}^{\mathsf{T}}(\dot{\mathbf{x}} + \lambda\mathbf{x}) + \mathbf{p} \\ \mathbf{V} &\leq \mathbf{T} \end{aligned} \quad \Leftrightarrow \quad \begin{aligned} \dot{\mathbf{V}} &= -\lambda\mathbf{V} + \mathbf{D}^{\mathsf{T}}(\dot{\mathbf{x}} + \lambda\mathbf{x}) \\ \mathbf{V} &\leq \mathbf{T} - h * \mathbf{p} \ \text{ with } \ h(t) = \Theta(t)e^{-\lambda t}. \end{aligned} \tag{15}$$

Here, $\mathbf{p}(t)$ denotes the vector of current perturbations, and $h * \mathbf{p}$ denotes a convolution of the perturbation currents with an exponential kernel, $h(t)$. Note that moving the perturbation onto the threshold does not change the spiking behavior of the neuron. *Appendix 1—table 1* includes the range of perturbations used throughout this manuscript.

### Voltage noise

We implement voltage noise as an extra random current on the voltage dynamics. This extra current follows a Wiener process scaled by $\sigma_V$, which denotes the standard deviation of the noise process with Gaussian increments (see *Equation 8*). In the absence of recurrence,

$$dV_j(t) = -\lambda V_j(t)\, dt + \nu(t)\, \sqrt{dt}, \qquad \nu \sim \mathcal{N}_M(0, \sigma_V), \tag{16}$$

so that the leaky integration with time constant $\lambda$ biases the random walk of the thresholds back towards their default values. For stationary inputs, the thresholds therefore follow an Ornstein-Uhlenbeck process.

### Synaptic perturbations

We perturb synapses between different neurons ($i \neq j$) by a multiplicative noise term

$$\Omega_{i,j} \leftarrow \Omega_{i,j} * (1 - \delta_\Omega)^{u_{i,j}}, \tag{17}$$

where $u_{i,j} \sim \mathcal{U}(-1,1)$. Here, the parameter $\delta_\Omega$ is the maximum weight change in percentage of each synapse, which in *Figure 6G* is referred to as maximal synaptic scaling.

## Synaptic delays

We implement delayed recurrent connections with the same constant delay length $\theta \geq 0$ for all pairs of neurons. Regardless of whether or not lateral excitation and inhibition are delayed in this way, the self-reset of a neuron onto itself remains instantaneous. *Equation 3* thus becomes

$$V_i = \mathbf{D}_i^\top \boldsymbol{x} - \sum_{k=1}^N \mathbf{D}_i^\top \mathbf{D}_k \left( r_k(t) \cdot \delta_{ik} + r_k(t-\theta) \cdot (1-\delta_{ik}) \right), \tag{18}$$

where $\delta_{ik}$ is Kronecker's delta. We assume that the decoder readout is equally delayed.

## Parameter choices

The spiking networks presented here depend on several parameters:

1. The number of neurons in the network, $N$.
2. The number of signals fed into the network, $M$, also called the dimensionality of the signal.
3. The $M \times N$ matrix of decoding weights, $D_{ik}$, where each column $\mathbf{D}_k$, corresponds to the decoding weights of one neuron.
4. The inverse time constant of the exponential decay of the readout, $\lambda$.
5. The threshold (or error tolerances) of the neurons, $T$.
6. The refractory period, $\tau_{\mathrm{ref}}$.
7. The current noise, $\sigma_V$.

These parameters fully define both the dynamics and architecture – in terms of feedforward and recurrent connectivity – of the networks, as well as the geometry of the bounding box. We studied networks with various number of neurons $N$ and input dimensionality $M$. The decoding weights of each neuron were drawn from an $M$-dimensional standard normal distribution,

$$\mathbf{D}_j \sim \mathcal{N}(0, \boldsymbol{I}), \tag{19}$$

and then normalized,

$$\mathbf{D}_j \leftarrow \mathbf{D}_j / \|\mathbf{D}_j\|_2, \tag{20}$$

such that each neuronal decoding vector is of length 1. We then did a sweep on the remaining parameters ($\lambda$, $T$, $\tau_{\mathrm{ref}}$, $\sigma_V$), to narrow down the range of parameters that roughly matches key observational constraints, such as low median firing rates ($\sim 5$ Hz), as found in cortex (*Hromádka et al., 2008*; *Wohrer et al., 2013*; *Figure 5B*), and coefficients of variation of interspike intervals close to one for each neuron, corresponding to Poisson-like spike statistics (*Figure 5C*). *Appendix 1—table 1* displays the range of parameters used to simulate baseline and perturbed networks.

## Input signal

We used two different types of inputs throughout our simulations. The results shown in *Figure 4* and *Appendix 1—figure 5* are for a circular, 2-dimensional signal,

$$\mathbf{x}(t) = \left( a\sin(\omega t), a\cos(\omega t) \right)^\top, \tag{21}$$

with constant amplitude $a$ and constant frequency $\omega$.

For all other simulations shown in figure panels, the input signal was a constant signal with additive noise. More precisely, for each trial, we sampled a single point in input space from an $M$-dimensional Gaussian distribution,

$$\mathbf{x}_0 \sim \mathcal{N}\left( \mathbf{0}, \sigma_x^2 \mathbf{I} \right). \tag{22}$$

The input signal ramps linearly from zero to this point $\boldsymbol{x}_0$ during the first 400ms. For the rest of the trial, the input to the neurons is set to slowly vary around this chosen input vector. To generate the slow variability, we sampled from an $M$-dimensional Gaussian distribution as many times as there were time steps in the rest of the trial; we then twice-filtered the samples with a moving average window of

1s for each dimension of $\mathbf{x}$, and for each dimension of $\mathbf{x}$ and across time, we normalized the individual slow variabilities to not exceed $\eta_x = 0.5$ in magnitude. This procedure was chosen to mimic experimental trial-to-trial noise.

## Metrics and network benchmarking

To compare the behavior of our networks under baseline conditions to those under the different perturbations, we need reliable measures of both coding accuracy and firing statistics. Below, we describe the measures used in this study.

### Distributions of firing rates and coefficients of variation

We measured the time-averaged firing rate for a given neuron by dividing the total number of spikes by the total duration of a trial. The coefficient of variation (CV) of a single spike train is computed as the ratio of the standard deviation of the interspike intervals (ISI) to their mean

$$\text{CV} = \frac{\sigma_{\text{ISI}}}{\mu_{\text{ISI}}}. \tag{23}$$

We recorded the full distributions of both the firing rates and CVs for a given network, pooling across neurons and different trials.

### Network performance

When our aim was to compare the relative network performance with and without the different perturbations, we opted to use a simple Euclidean distance or L2 norm to measure the average error of each network:

$$E = \langle \|\mathbf{x}(t) - \hat{\mathbf{x}}(t)\|_2 \rangle_t. \tag{24}$$

To compute the relative performance, we divided the error of the perturbed network by the error of the equivalent, unperturbed network using the formula

$$P = \frac{E_{\text{perturbed}} - E_{\text{dead}}}{E_{\text{reference}} - E_{\text{dead}}}, \tag{25}$$

where $E_{\text{dead}}$ is the error of a non-functional or dead network ($\hat{\mathbf{x}}(t) = 0$ or $E_{\text{dead}} = \langle \|\mathbf{x}(t)\| \rangle_t$). We included this case to provide a baseline, worst-case scenario.

A key limitation with most error measures is that they scale in various ways with dimensionality. This becomes an issue in *Figure 5* as this hinders the comparison of errors across different signal dimensionalities. For this particular case, we chose to measure the coding errors in a dimensionality-independent way by pooling together the errors in each individual signal component, $|x_i - \hat{x}_i|$. We can then compute the median of this aggregated distribution in order to consistently compare the performance of these networks across different signal dimensionalities.

### Excitation-inhibition balance

In order to compute the EI balance of a given neuron $j$, we divided the total synaptic input throughout a given trial into its positive ($C_j^+$) and negative ($C_j^-$) components

$$C_j^+ = \int dt \left( \max \left( \mathbf{D}_j^\mathsf{T} \left( \lambda \mathbf{x} + \dot{\mathbf{x}} \right), 0 \right) + \sum_{k \neq j} \max \left( -\mathbf{D}_j^\mathsf{T} \mathbf{D}_k s_k, 0 \right) \right) \tag{26}$$

$$C_j^- = \int dt \left( -\min \left( \mathbf{D}_j^\mathsf{T} \left( \lambda \mathbf{x} + \dot{\mathbf{x}} \right), 0 \right) - \sum_{k \neq j} \min \left( -\mathbf{D}_j^\mathsf{T} \mathbf{D}_k s_k, 0 \right) \right). \tag{27}$$

The normalized E-I difference $b_j$ of a neuron $j$ was then computed as

$$b_j = \frac{C_j^+ - C_j^-}{C_j^+ + C_j^-}. \tag{28}$$

In other words, $b_j = 0$ if a neuron is perfectly balanced, $0 < b_j \leq 1$ if a neuron receives more excitation than inhibition and $-1 \leq b_j < 0$ if a neuron receives more inhibition than excitation.

### Benchmarking

To fully compare the behavior of the networks under baseline conditions to those under the different perturbations, we adopted the following benchmarking procedure: each simulated trial with a perturbation is compared to an otherwise identical trial without perturbation. For each trial, we generated a new network with a different random distribution of decoding weights, random input signal, and random voltage noise. These parameters were used for both the perturbed and unperturbed trial. We then applied our $M$-dimensional input signal $\mathbf{x}$ as described above, and recorded coding error and spiking statistics for both perturbed and unperturbed trial. This procedure was repeated multiple times ($N_{\text{trials}} \geq 20$), each repetition resulting in different network connectivity, inputs, and injected current noise, and each pair of trials returning one performance value as defined above.

We choose this benchmarking procedure to sample the space of input signals in an unbiased way. This ensures that network performance is not accidentally dominated by a perfect match, or mismatch, between the fixed decoding weights and a given random input. Particularly bad mismatches may still lead to high decoding errors, but because our error measure considers the median response, these extremes do not bias our benchmarking procedure.

### Number of simulations

*Figure 1D* shows a single trial. *Figure 5* shows a total of 29,400 trials. *Figure 6E and F* show 16,830 pairs of trials, and *Appendix 1—figure 4* shows 4996 pairs. Panels *Figure 6G and H* consist of 840 trials each. *Figure 7G* show 18,000 pairs of trials, or 200 pairs per data point, and *Appendix 1—figure 5* shows 1 perturbed trial per row.

## Numerical implementation

We numerically solve the differential equations (*Equation 8*) describing the temporal evolution of membrane voltage by the forward Euler-Maruyama method. Because of finite simulation time steps, more than one neural threshold may be crossed during the same step, and more than one neuron may thus be eligible to spike. This problem can be circumvented by decreasing the time step, which, however, increases simulation time. To avoid this tradeoff, we essentially slow down time whenever multiple neurons crossed threshold (Appendix 1—algorithm 1). Note that when considering finite delays $\theta$, delayed lateral recurrence arrives only at the end of each time step (Appendix 1—algorithm 2).

We implemented these methods in both MATLAB and Python, and both sets of code can be used interchangeably. Our code for simulation, analysis and figure generation, as well as sample data files can be found at https://github.com/machenslab/boundingbox (copy archived at swh:1:rev:d9ce2c-f52e833ecf67dccc796bd8c9dc505f2e00, *Calaim, 2022*), under a Creative Commons CC BY-NC-SA 4.0 license.

## Iterative adaptation of parameters to avoid ping-pong

In networks with delays, we can avoid ping-pong either by increasing box size or by removing a number of strongest excitatory connections. In both cases, we compute the minimum required value offline using an iterative procedure (Appendix 1—algorithm 3). Note that trials must be sufficiently long to avoid false-negative reports of ping-pong.

## Movie visualization

All movies (*Videos 1 and 2*) were produced in Python, with the exception of the three-dimensional visualization of a polytope, for which we used the *bensolve* toolbox for MATLAB (*Löhne and Weißing, 2017*).

## FORCE-learning rate network and perturbations

We trained a recurrent network of 1000 rate units using FORCE learning (*Sussillo and Abbott, 2009*) in the absence of any perturbation. The network dynamics are described by the following system of differential equations:

$$\tau\frac{d\mathbf{x}}{dt} = -\mathbf{x} + g\mathbf{J}\mathbf{r} + \mathbf{J_z}\mathbf{z} \tag{29}$$

where $\mathbf{r} = \tanh(\mathbf{x})$ corresponds to the firing rates, and where $\tau = 10ms$, $g = 1.5$, and $\mathbf{J}$ is a sparse random matrix whose elements are zero with probability $1 - p$. Each nonzero element is drawn independently from a Gaussian distribution with zero mean and variance equal to $(1000p)^{-1}$. The entries of the matrix $\mathbf{J}_z$ are uniformly distributed from –1 to 1.

We then applied one of three perturbations to the fully trained network: neuron death, rate noise, or synaptic perturbation. We emulated neural loss by setting the respective neural activities to zero, i.e. $x_i = 0$. The rate noise perturbation was simulated by injecting white noise within the input-output non-linearity and its magnitude was chosen so that fluctuations on the network activities were of the same order of magnitude as the ones simulated for coordinated spiking networks. Finally, we simulated synaptic perturbations following the same procedure and magnitude as for the coordinated spiking networks, i.e., each element of the recurrent connectivity matrix was changed randomly up to 2.5% of its value.

## Acknowledgements

We thank Alfonso Renart for helpful discussions and comments on the manuscript. This work was supported by the Fundação para a Ciência e a Tecnologia (project FCT-PTDC/BIA-OUT/32077/2017-IC&DT-LISBOA-01-0145-FEDER) and by the Simons Foundation (Simons Collaboration on the Global Brain #543009).

## Additional information

### Funding

| Funder | Grant reference number | Author |
|---|---|---|
| Fundação para a Ciência e a Tecnologia | FCT-PTDC/BIA-OUT/32077/2017-IC&DT-LISBOA-01-0145-FEDER | Christian K Machens |
| Simons Foundation | 543009 | Christian K Machens |
| Fundação para a Ciência e a Tecnologia | SFRH / BD / 52217 / 2013 | Nuno Calaim |

The funders had no role in study design, data collection and interpretation, or the decision to submit the work for publication.

### Author contributions

Nuno Calaim, Florian A Dehmelt, Conceptualization, Data curation, Formal analysis, Investigation, Methodology, Project administration, Software, Validation, Visualization, Writing – original draft, Writing – review and editing; Pedro J Gonçalves, Project administration, Supervision, Writing – original draft, Writing – review and editing; Christian K Machens, Conceptualization, Funding acquisition, Methodology, Project administration, Resources, Software, Supervision, Visualization, Writing – original draft, Writing – review and editing, Formal analysis, Investigation, Validation

### Author ORCIDs

Nuno Calaim http://orcid.org/0000-0003-0317-3276
Florian A Dehmelt http://orcid.org/0000-0001-6135-4652
Pedro J Gonçalves http://orcid.org/0000-0002-6987-4836
Christian K Machens http://orcid.org/0000-0003-1717-1562

### Decision letter and Author response

Decision letter https://doi.org/10.7554/eLife.73276.sa1
Author response https://doi.org/10.7554/eLife.73276.sa2

## Additional files

**Supplementary files**
• Transparent reporting form

**Data availability**
The current manuscript is a computational study, so no data have been generated for this manuscript. Modelling code is uploaded on https://github.com/machenslab/boundingbox, (copy archived at swh:1:rev:d9ce2cf52e833ecf67dccc796bd8c9dc505f2e00).

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

## Appendix 1

**Appendix 1—algorithm 1.** Numerical implementation of a general network with voltage noise $\sigma_V$ and refractory period $\tau_{\text{ref}}$.

$K \leftarrow \{k \mid k \in \mathbb{N} : 1 \leq k \leq N\}$ //all neurons initialise
**for** $t = 0$ to $t_{\text{max}}$ in steps $\Delta t$ **do**

 $R \leftarrow \left\{ k \mid k \in K : t - \arg\max_{t' < t} \left( s_k(t') = 1 \right) < \tau_{\text{ref}} \right\}$ //in refraction

 $C \leftarrow \left\{ k \mid k \in K \backslash R : V_k(t) > T_k(t) \right\}$ //spike candidates

 **while** $C \neq \emptyset$ **do**
 $w \leftarrow \arg\max_{k \in C} \left( V_k(t) - T_k(t) \right)$ // furthest above threshold
 $s_w(t) \leftarrow 1$ //spike
 $\boldsymbol{V}(t) \leftarrow \boldsymbol{V}(t) - \boldsymbol{D}^{\mathsf{T}} \boldsymbol{D}_w$ //instant recurrence
 $R \leftarrow R \cup \{w\}$ // refraction
 $C \leftarrow \left\{ k \mid k \in K \backslash R : V_k(t) > T_k(t) \right\}$ // spike candidates
 **end**
 sample $\boldsymbol{\eta}(t) \sim \mathcal{N}(\boldsymbol{0}, \sigma_V \boldsymbol{I})$
 $\boldsymbol{V}(t + \Delta t) \leftarrow \boldsymbol{V}(t) + \Delta t \left( -\lambda \boldsymbol{V}(t) + \lambda \boldsymbol{D} \boldsymbol{x}(t) \right) + \sqrt{\Delta t} \, \boldsymbol{\eta}(t)$
**end**

**Appendix 1—algorithm 2.** Numerical implementation of a general network with finite delays $\theta$, refractory period $\tau_{\text{ref}}$, current noise $\sigma_V$, time-varying synaptic noise $\Delta\mathbf{\Omega}(t)$ and time-varying optogenetic currents $\boldsymbol{p}(t)$.

$K \leftarrow \left\{ k \mid k \in \mathbb{N} : 1 \leq k \leq N \right\}$ // all neurons initialise

$V_k(0) \; \forall \, k \in K$

$\mathbf{\Omega} = \boldsymbol{D}^{\mathsf{T}} \boldsymbol{D}$ // standard recurrent matrix

**if** $\theta > 0$ **then**

 $\mathbf{\Omega}^f = \text{diag}(\mathbf{\Omega})$ // instant self-reset vector

 $\mathbf{\Omega}^\theta = \mathbf{\Omega} - \text{diag}(\mathbf{\Omega}^f)$ // delayed recurrence matrix

**end**

**for** $t = 0$ to $t_{\max}$ in steps $\Delta t$ **do**

 sample $\mathbf{\Omega}^*(t) \leftarrow \mathbf{\Omega} + \Delta\mathbf{\Omega}(t)$ // synaptic noise

 **if** $\theta > 0$ **then**

 $\mathbf{\Omega}^f = \text{diag}(\mathbf{\Omega}^*(t))$ // instant self-reset vector

 $\mathbf{\Omega}^\theta = \mathbf{\Omega}^*(t) - \text{diag}(\mathbf{\Omega}^f)$ // delayed recurrence matrix

 **end**

 $R \leftarrow \left\{ k \mid k \in K : t - \arg\max_{t' < t} \left( s_k(t') = 1 \right) < \tau_{\text{ref}} \right\}$ // in refraction

 $C \leftarrow \left\{ k \mid k \in K \backslash R : V_k(t) > T_k(t) \right\}$ // spike candidates

 **while** $C \neq \emptyset$ **do**

 $w \leftarrow \arg\max_{k \in C} \left( V_k(t) - T_k(t) \right)$ // furthest above threshold

 $s_w(t) \leftarrow 1$ // spike

 **if** $\theta > 0$ **then**

 $V_w(t) \leftarrow V_w(t) - \mathbf{\Omega}_w^f$ // instant self-reset

 **else**

 $\boldsymbol{V}(t) \leftarrow \boldsymbol{V}(t) - \mathbf{\Omega}_w^*$ // instant recurrence

 **end**

 $R \leftarrow R \cup \{w\}$ // refraction

 $C \leftarrow \left\{ k \mid k \in K \backslash R : V_k(t) > T_k(t) \right\}$ // spike candidates

 **end**

 $\Delta\boldsymbol{V} = \Delta t \left( -\lambda \boldsymbol{V}(t) + \lambda \boldsymbol{D} \boldsymbol{x}(t) \right)$ // dynamics unperturbed network

 sample $\boldsymbol{\eta}(t) \sim \mathcal{N}(\boldsymbol{0}, \sigma_V \boldsymbol{I})$

 $\Delta\boldsymbol{V} \leftarrow \Delta\boldsymbol{V} + \sqrt{\Delta t} \, \boldsymbol{\eta}(t)$ // current noise

 $\Delta\boldsymbol{V} \leftarrow \Delta\boldsymbol{V} + \Delta t \, \boldsymbol{p}(t)$ // optogenetic currents

 **if** $\theta > 0$ **then**

 $\Delta\boldsymbol{V} \leftarrow \Delta\boldsymbol{V} - \mathbf{\Omega}^\theta \boldsymbol{s}(t + \Delta t - \theta)$ // delayed recurrence

 **end**

 $\boldsymbol{V}(t + \Delta t) \leftarrow \boldsymbol{V}(t) + \Delta\boldsymbol{V}$

**end**

**Appendix 1—algorithm 3.** Numerical search for the "safe width" of a bounding box, avoiding ping-pong. Typical parameters are $T_{\min} = 0.55$, $\alpha = 1.5$, $\beta = 0.95$, $\gamma = 0.1$, $\epsilon = 0.05 \cdot 2\theta$, $N = 100$. In each trial, all neurons $j$ have the same threshold $T_j$, and the box is thus widened or narrowed symmetrically.

---

| | |
|---|---|
| initialise $T \leftarrow T_{\min} > 0$ | // current box width |
| initialise $T^* \leftarrow 0$ | // best box width so far |
| initialise $k \leftarrow 0$ | // trial counter |
| while $k < K$ do | |
| $\quad k \leftarrow k + 1$ | |
| $\quad$ simulate network with $N$ neurons and box width $T$ | |
| $\quad$ for $1 < j \leq N$ do | |
| $\qquad \Theta_j \leftarrow \{t \mid s_j(t) = 1\}$ | // spike times |
| $\qquad S_j \leftarrow \{t - t' \mid t, t' \in \Theta_j \ \wedge \ t = \arg\min_x (x > t')\}$ | // intervals |
| $\quad$ end | |
| $\quad S \leftarrow \bigcup_{j=1}^N S_j$ | // pool interspike intervals |
| $\quad A \leftarrow \{a \in S \mid 2\theta - \epsilon < a < 2\theta + \epsilon\}$ | // SISIs near double-delay |
| $\quad P \leftarrow \dfrac{|A|}{|S|} > \gamma$ | // Boolean: ping-pong present? |
| $\quad$ if $P$ then | |
| $\qquad$ if $w^* > 0$ then | |
| $\qquad\quad w \leftarrow T^*$ | // use previous estimate... |
| $\qquad\quad k \leftarrow K$ | //...and quit |
| $\qquad$ else | |
| $\qquad\quad T \leftarrow \alpha T$ | // increase box size |
| $\qquad\quad k \leftarrow 0$ | // restart trial counter |
| $\qquad$ end | |
| $\quad$ else if $k = N$ then | |
| $\qquad T^* \leftarrow w$ | // update best estimate |
| $\qquad T \leftarrow \beta T$ | // slightly decrease box size |
| $\qquad k \leftarrow 0$ | // restart trial counter |
| $\quad$ end | |
| end | |

---

**Appendix 1—table 1.** Network parameter values.

| | Variable (Unit) | baseline value | value range |
|---|---|---|---|
| $N$ | network size | | [2, 5,000] |
| $M$ | signal dimensions | | [1, 100] |
| $\rho$ | network redundancy N/M | | [2, 50] |
| $\|\boldsymbol{D}_i\|_2$ | decoder norms | 1 | |
| $\frac{1}{\lambda}$ | decoder time constant (ms) | 10 | |
| $T_i$ | threshold (a.u.) | 0.55 | [0.5, 1.55*] |
| $t_{\max}$ | trial duration (s) | 5 | |
| $\Delta t$ | simulation time step (ms) | 0.1 | [0.01 0.1] |
| $\sigma_x$ | standard deviation of each signal component | 3 | |
| $\eta_x$ | signal noise | 0.5 | |
| $\tau_{\mathrm{ref}}$ | refractory period (ms) | 2 | [0, 10] |
| $V_{i,\mathrm{reset}}$ | reset (a.u.) | 1.014 | [1, 1.5] |
| $\sigma_V$ | current noise (a.u.) | 0.5 | [0, 3] |
| $\delta_\Omega$ | synaptic scaling/noise | 0 | [0, 0.2] |
| $\theta$ | recurrent delay (ms) | 0 | [0, 2] |

*To counteract synaptic delays as in **Figure 7**, thresholds $T > 1.55$ were also used.

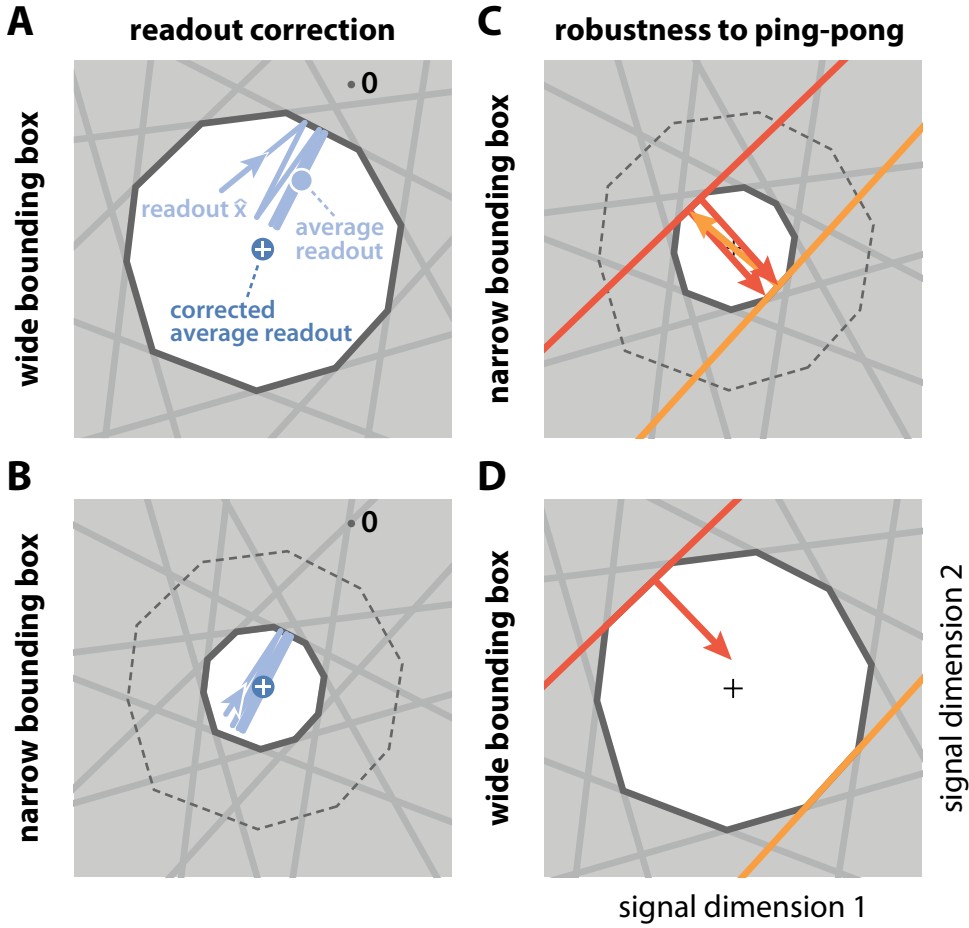

**Appendix 1—figure 1.** Wide and narrow boxes, ping-pong, and readout correction. (**A**) Wide box. Upon each spike, the readout (light blue) jumps into the box, but without reaching its opposite end, and then decays back to the border of the box. As a consequence, the readout fluctuates around a mean readout vector (light blue, solid circle) that is shorter than the input signal vector (white cross). The coding error therefore has two components, one corresponding to the readout fluctuations, and one to the systematic bias. This bias can be corrected for (Methods, 'Readout biases and corrections'; mean shown as dark blue solid circle). (**B**) Narrow box. When the box diameter is the size of the decoding vectors, the systematic bias vanishes, and both corrected and uncorrected readout are virtually identical. (**C**) Ping-pong. In narrow boxes, a spike will take the readout all the way across the box, increasing the likelihood that even a small amount of noise will trigger unwanted 'pong' spikes (orange arrow) in the opposite direction, followed by further 'ping' spikes in the original direction (red arrows). Such extended barrages lead to excessive increases in firing rates and are referred to as the 'ping-pong' effect. (**D**) Avoiding ping-pong. In wide boxes, when the readout hits one of the bounds (red line), the resulting spike (red arrow) will take it well inside the box. Even in the presence of e.g. voltage or threshold noise, this is unlikely to result in additional spikes in the opposite direction. (However, note that at high dimensionality or very low redundancy, the complex geometry of the bounding box can sometimes result in a finite number of instantaneous compensatory spikes).

$M = 2, \rho = 3$   $M = 50, \rho = 20$

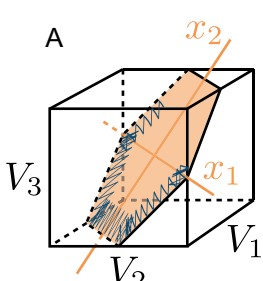 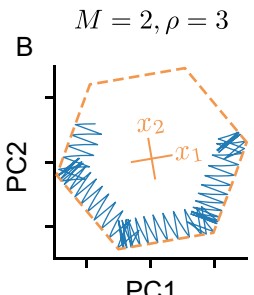 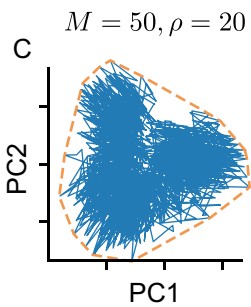

**Appendix 1—figure 2.** Physical manifestation of the bounding box in the network's voltage space. (**A**) A (noise-free) network with $N = 6$ neurons tracking a two-dimensional signal. We assume that the neuron's decoding vectors are regularly spaced. In this case, the voltages of neurons with opposite decoding vectors ($\boldsymbol{D}_i = -\boldsymbol{D}_{i+3}, i \in \{1, 2, 3\}$) can be collapsed into single dimensions (since $V_i = -V_{i+3}$). In turn, we can plot the six-dimensional voltage space in three dimensions, as done here. The inside of the cube corresponds to the subthreshold voltages of the neurons, and the faces of the cube to the six neural thresholds. The network's voltage trajectory is shown in blue and lives in a two-dimensional subspace (orange). The limits of this subspace, given by the neuron's thresholds, delineate the (hexagonal) bounding box. (**B**) We apply Principal Component Analysis to the original six-dimensional voltage traces to uncover that the system only spans a lower two-dimensional subspace which shows the original bounding box. (**C**) Same as B, but for a high-dimensional and high-redundancy system ($M = 50$, $N = 1000$, $\rho = 20$). In this case, the first two principal components only provide a projection of the original bounding box, and the voltage trajectories are unlikely to exactly trace out the projection's boundaries.

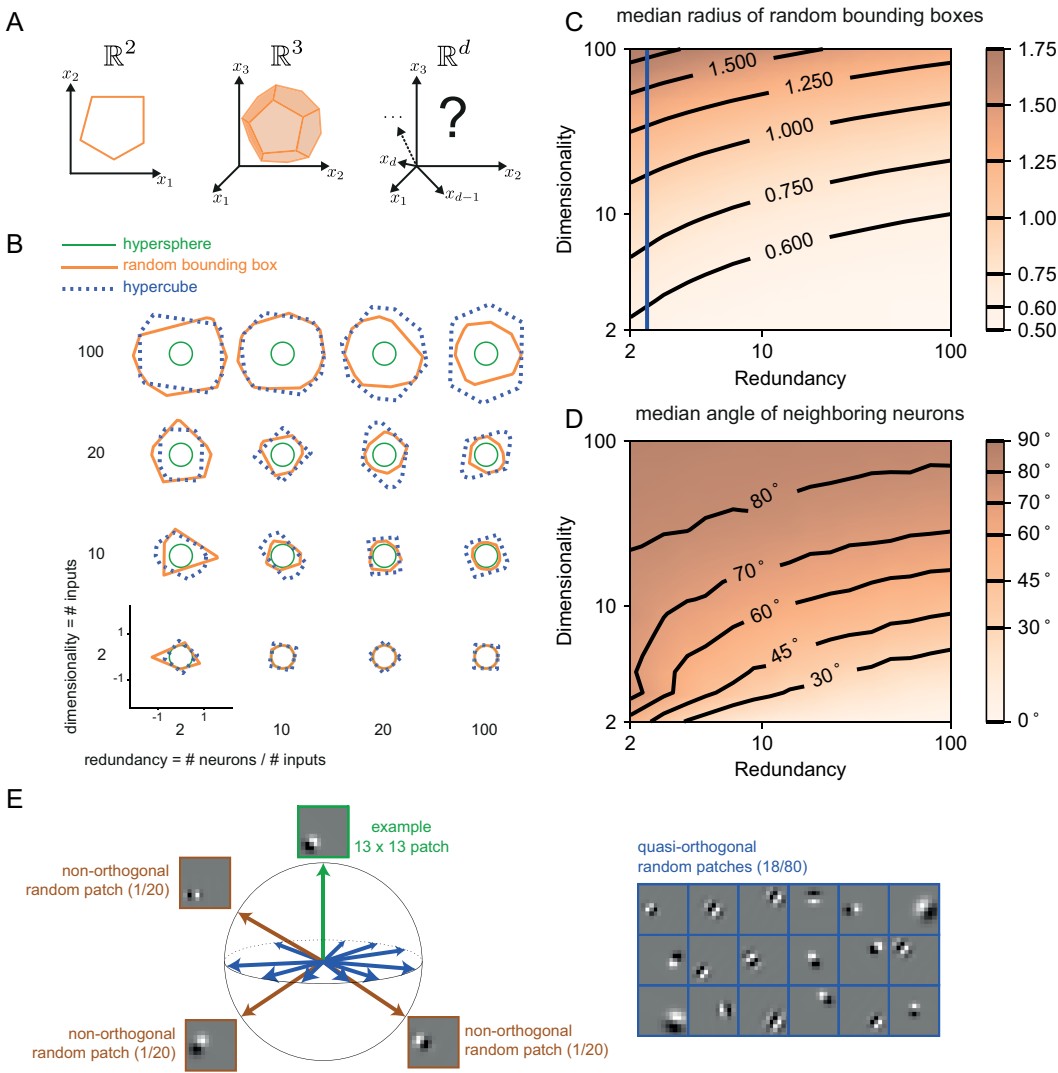

**Appendix 1—figure 3.** The geometry of the bounding box changes with input dimensionality and redundancy. (**A**) In networks tracking two-dimensional signals, the bounding box is geometrically depicted as a polygon with as many sides as the number of neurons. For three dimensional systems, the bounding box corresponds to a polyhedron. For four or more dimensions, the corresponding bounding boxes are mathematically described as convex polytopes, but their visualization is hard (see Materials and methods, 'Geometry of high-dimensional bounding boxes'). (**B**) Example two-dimensional cuts of bounding boxes (orange) for a given network size and space dimensionality. Cuts for a hypersphere (green) and a hypercube (dashed blue) are shown for comparison. For low dimensionality, high redundancy bounding boxes are similar to hyperspheres whereas for high dimensionality they are more similar to hypercubes. (**C**) Median radius of bounding boxes as a function of dimensionality and redundancy. The blue line illustrates the average radius of a hypercube (thresholds of individual neurons are here set at T=0.5). (**D**) Median angle between neighbouring neurons, i.e., neurons that share an 'edge' in the bounding box. Neighbouring neurons in high dimensional signal spaces are almost orthogonal to each other (**E**) Random 13 × 13 Gabor Patches representing the readout weights of neurons in a high dimensional space. Most Gabor patches are quasi-orthogonal to each other (angles within $90 \pm 5°$). Some neurons have overlapping receptive fields and non-orthogonal orientations.

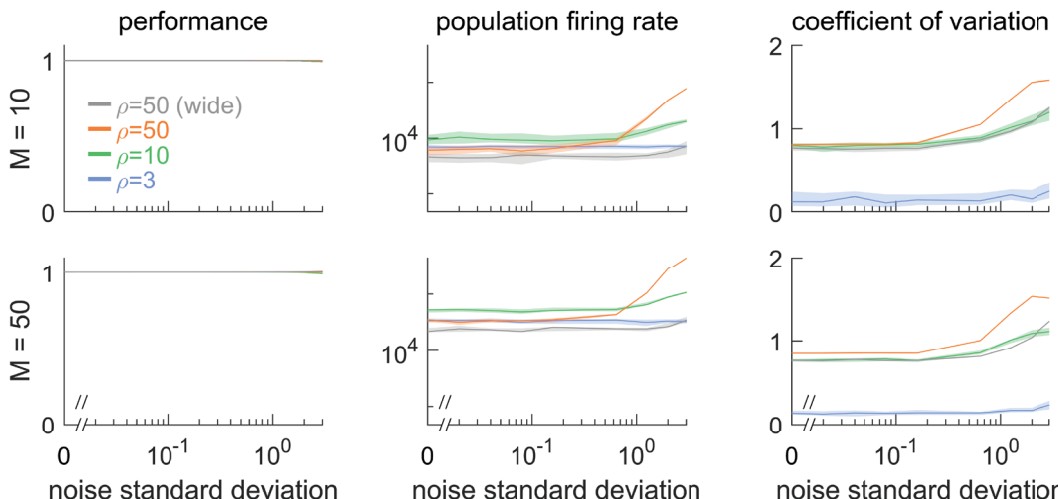

**Appendix 1—figure 4.** Robustness to noise for different signal dimensionalities ($M = 10$ and $M = 50$) and different redundancies $\rho$. (Left column) Network performance relative to an identical reference network without noise. Different curves lie on top of each other. (Central column) Population firing rate. (Right column) Coefficient of variation of the interspike intervals, averaged across neurons. Overall, dimensionality does not qualitatively affect robustness to noise. Threshold is $T = 0.55$ by default, unless labeled 'wide', which corresponds to an expanded threshold of $T = 1.0$. Lines show medians, and shaded regions indicate interquartile ranges.

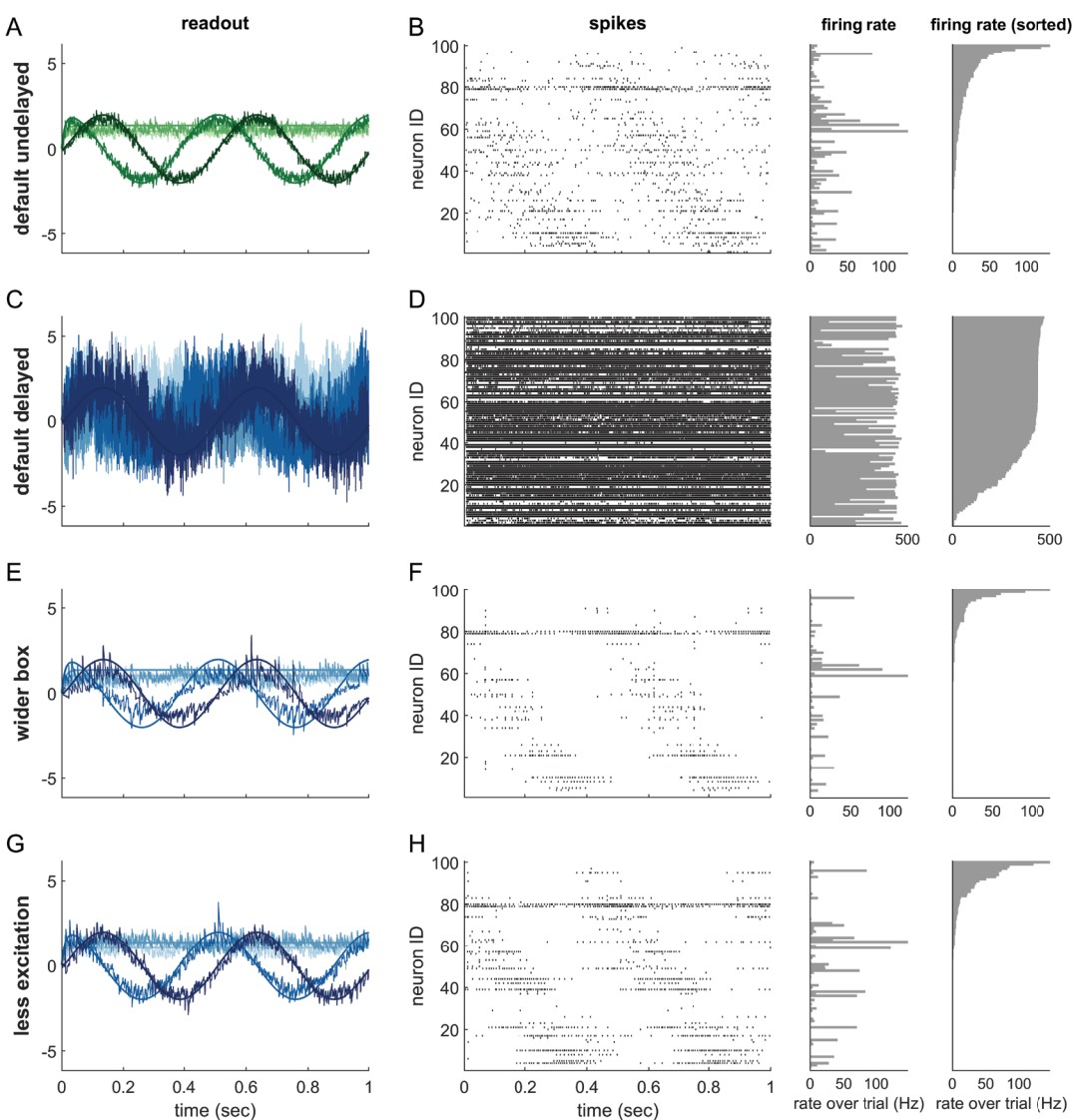

**Appendix 1—figure 5.** Single trials of delayed and undelayed networks for intermediate dimensionalities (number of input signals $M = 20$, redundancy $\rho = 5$). The input signals are a sine and cosine along the first two dimensions, and constant along the remaining dimensions. (**A,B**) Undelayed, fully connected network with a default box ($T = 0.55$), (**C,D**) Delayed, fully connected network with a default box, (**E,F**) delayed fully connected network with optimally widened box, (**G,H**) delayed network with default box and optimally reduced excitation. (**C–H**) Delay is $\theta = 1$ms. Panels (**A,C,E,G**) show the readout in each of the first four signal dimensions as a separate line. Dimensions 5–20 are hidden to avoid clutter. Panels (**B,D,F,H**) show corresponding spike-time raster plots (left) and trial-averaged single-neuron firing rates (centre), as well as the same rates ordered from largest to smallest (right).

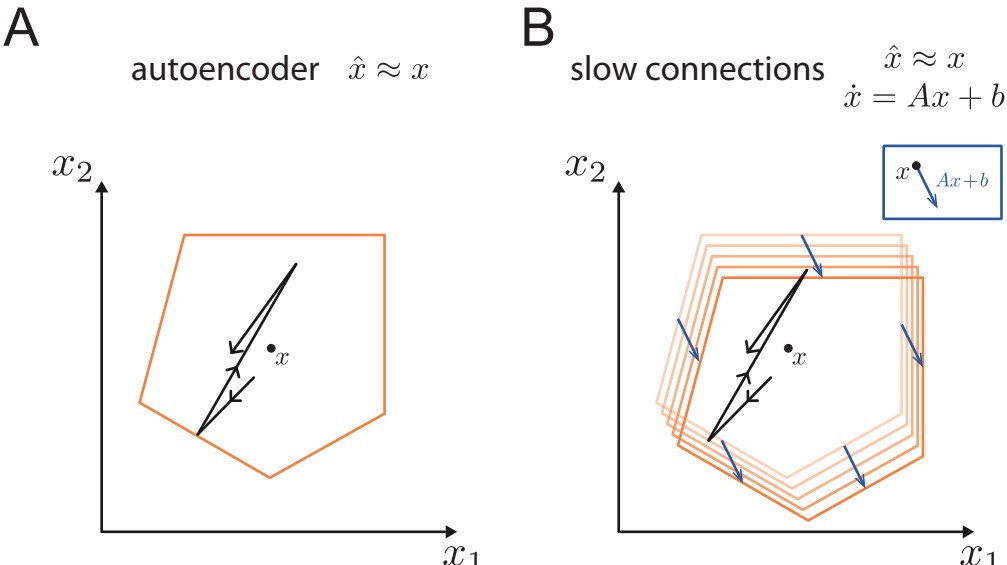

**Appendix 1—figure 6.** Generalisation of the bounding box. (**A**) In a simple autoencoder, the input is directly fed into the network. During a spike, the bounding box maintains its overall shape due to the network's fast recurrent connectivity. (**B**) When we add dynamics, the resulting networks have the same fast recurrent connectivity matrix as the auto-encoder networks, and this fast recurrency maintains the bounding box during a spike. Additionally, the networks have a slow, recurrent connectivity matrix. We can visualize the effect of this slow recurrent connectivity by treating it as a perturbation, similarly to the other perturbations discussed in the paper. The effect of the slow connectivities is then to move the bounds of the neurons according to the evolution of the dynamical system. Perturbations for which the autoencoder is robust, i.e., for which the readout error is kept within normal range, will therefore not affect the slow dynamics.

