## [Editor Report]

The article introduces a geometrical interpretation for the dynamics and function of certain spiking networks, based on the earlier work of Machens and Deneve. Given that spiking networks are notoriously hard to understand, the approach could prove useful for many computational neuroscientists. Here, that visualization tool serves to assess how fragile the network is to perturbation of its parameters, such as neuronal death, or spurious noise in excitation and inhibition.

---

## [Decision Letter]

**Decision letter after peer review:**

Thank you for submitting your article "Robustness in spiking networks: a geometric perspective" for consideration by *eLife*. Your article has been reviewed by 3 peer reviewers, including Markus Meister as Reviewing Editor and Reviewer #3, and the evaluation has been overseen by Michael Frank as the Senior Editor. The following individual involved in review of your submission has agreed to reveal their identity: Brent Doiron (Reviewer #2).

Essential revisions:

Domain of applicability:

1. The authors analyze a very specific network, with carefully tuned feedback designed for a specific cost function (Equation 3). Do such tuned networks exist in Nature? The reader would benefit from knowing what the domain of applicability really is.

2. How would the framework generalize to settings other than an auto-encoder, e.g., for the dynamical systems approach in Boerlin et al.?

Consistency of the bounding box picture:

3. A central assumption of the framework is a coherence between the readout from the network and the dynamics and connectivity of the network. This allows the bounding box to follow the input vector around, so one can easily assess decoding errors. However, it seems that some of the perturbations break the assumptions that lead to the bounding box. For example, exciting a single neuron decreases its threshold and thereby shrinks the bounding box (Figure 3D). As the bounding box limits the error in the readout, this should mean that the error decreases. However, the opposite is the case (Figure 4A). Presumably the discrepancy arises from changing the threshold without changing the readout at the same time, so the bounding box loses its meaning. By contrast the loss of a neuron, or its inhibition, leaves the remaining network adjusted properly, so the bounding box retains its meaning. The authors should address this question, and whether the bounding box loses its meaning under some perturbations and not others.

4. A related question applies to delays. Where exactly are the delays? In the readout and then the network is optimised for that? Or is the network the optimal one assuming no delays and then delays are added in the recurrent connections and/or the readout? Clarifying that will make it easier for the reader to follow the argument.

5. In the section on perturbations of the reset potential: Is there again an asymmetry, e.g., do changes up or down in the optimal reset potential have very different consequences?

Robustness claims:

6. There are frequent comparisons between networks with few neurons and networks with many neurons. The idea that networks with more neurons are more robust seems kind of obvious. But the authors analyze a very specific network tuned for coordinated redundancy (Equation 3). How does that network compare to one with the same number of neurons but no coordinated redundancy? This would seem a more interesting comparison than varying the number of neurons.

7. Intuitively one might expect that the performance is fragile to changes in the recurrent coupling, since the decoding vector is tied to the feedforward and recurrent weights (Equation 10). Because of this the claimed robustness to perturbations in synaptic weight seems surprising. Is the 'synapse scaling factor' in Figure 6G,H the parameter \δ_{\Omega} in Equation 17? The way it is defined it is the maximal change, however most synapses are perturbed to a much smaller extent. Also, for \δ approaching 0.2 the performance only drops by 20% (Figure 6G). However, for such large \δ's the firing rates in the network seem unreasonably high (Figure 6H), suggesting it no longer performs efficiently. Overall it is hard to judge from the current presentation how robust the network is to synaptic perturbation. Perhaps the firing rates could be bounded, or the performance penalized for using very high rates?

Relation to plasticity:

8. Prior work (e.g. Bourdoukan 2012) proposed a Hebbian learning process that could tune up such a network automatically. If this plasticity is "always on" how would it respond to the various perturbations being considered? How might that contribute to robustness on a longer time scale? How does the size/shape of the bounding box depend on the parameters of the learning rule? Can one get looser and tighter boxes, as needed here for certain types of robustness?

---

## [Author Response]

Essential revisions:Domain of applicability:1. The authors analyze a very specific network, with carefully tuned feedback designed for a specific cost function (Equation 3). Do such tuned networks exist in Nature? The reader would benefit from knowing what the domain of applicability really is.

The reviewers are correct in that our networks require carefully tuned inhibitory feedback. We first notice that this is not a new proposal. For instance, in previous work we have shown that the loss functions we minimize are identical to classical sparse coding networks (Olshausen and Field, 1996). Indeed, the type of connectivity we find in our networks is exactly the same as that in sparse coding *rate* networks (compare e.g., Figure 10.6 in Dayan Abbott with Equation (8) in our paper). So we could say a potential applicability are sensory systems, such as primary visual cortex. Do such tuned networks really exist in nature? That is a more difficult question to answer. There is currently an ongoing debate on whether inhibitory tuning is precise or broad, and so we would say that the jury is still out (see e.g., Najafi et al., (2020) Neuron 105:165-79, for an argument that inhibitory tuning is precise or selective). We have sought to clarify these points in the discussion where we now write:

"In many respects, the bounding box is a "toy model" (borrowing the term from physics, in the sense of a deliberately abstract model), which we see mainly as a tool to conceptualize and highlight generic circuit mechanism, rather than an attempt to model any specific system. Nonetheless, it is worthwhile to point out that the bounding box is also a spiking version of classical sparse coding models of V1 [44]. Indeed, previous work has demonstrated that these spiking networks can explain various perturbation experiments in V1 [21]. So, besides shedding light on the robustness of coordinated spike codes, the bounding box can also be seen as a simple model of a sensory system."

2. How would the framework generalize to settings other than an auto-encoder, e.g., for the dynamical systems approach in Boerlin et al.?

The framework fully generalizes. In the case of an auto-encoder, the positions of bounding box faces are fully defined by the input. For the more general dynamical systems approach of Boerlin et al., there still is a bounding box that arises from the fast recurrent connectivity, and the position of its faces evolves in signal space according to the chosen dynamical system. See Supplementary Figure 6 for a graphical explanation of this comparison. We also note that the simulation in Figure 1D of the main paper is actually a dynamical system à la Boerlin et al., and not an auto-encoder.

To address these concerns, we added Supplementary Figure 6, and we introduced a new section at the end of the Results to clarify that our framework generalizes to Boerlin et al., Specifically, we wrote:

"The bounding box provides a useful tool even if we endow the networks with a set of slower connections to perform linear or non-linear computations [17, 42, 43]. Indeed, the simulation in Figure 1D used these slower connections to generate oscillatory dynamics (see Methods, section ’Generalisation of the bounding box IV’, for mathematical details). This extension to networks that generate persistent activity or dynamical patterns works because the mechanisms underlying the encoding of the signals into spike trains are decoupled from the mechanisms that generate the dynamics of the signals (or readouts). Accordingly, the extra currents generated by the slow recurrent connections can be seen as a perturbation of the bounding box thresholds. This perturbation shifts the bounding box in the space of readouts as illustrated in Supplementary Figure 6."

Consistency of the bounding box picture:3. A central assumption of the framework is a coherence between the readout from the network and the dynamics and connectivity of the network. This allows the bounding box to follow the input vector around, so one can easily assess decoding errors. However, it seems that some of the perturbations break the assumptions that lead to the bounding box. For example, exciting a single neuron decreases its threshold and thereby shrinks the bounding box (Figure 3D). As the bounding box limits the error in the readout, this should mean that the error decreases. However, the opposite is the case (Figure 4A). Presumably the discrepancy arises from changing the threshold without changing the readout at the same time, so the bounding box loses its meaning. By contrast the loss of a neuron, or its inhibition, leaves the remaining network adjusted properly, so the bounding box retains its meaning. The authors should address this question, and whether the bounding box loses its meaning under some perturbations and not others.

The reviewers are correct that the readout is a central pillar of our analysis. There is one important subtlety, though. For wide bounding boxes, i.e., for bounding boxes in which the readout jump caused by a spike does not transverse the whole box, we need to correct the readout for a bias (see Author response image 1, replotted from Supplementary Figure 1A). This bias-correction is done by rescaling the readout with a scalar. The bias correction achieves that the average rescaled readout will match the signal.

**Author response image 1. sa2fig1:** Wide box, and readout correction. Upon each spike, the readout (light blue) jumps into the box, but without reaching its opposite end, and then decays back to the border of the box. As a consequence, the readout fluctuates around a mean readout vector (light blue, solid circle) that is shorter than the input signal vector (white cross). The coding error therefore has two components, one corresponding to the readout fluctuations, and one to the systematic bias. This bias can be corrected for (Methods, ’Readout biases’), and we will sometimes work with the corrected readout (mean shown as dark blue solid circle).

The bounding box only limits the coding error of the actual readout, but not the rescaled, bias corrected readout. In other words, the bounding box only has meaning for the original readout, not the corrected readout. When we shrink the bounding box from one side, we decrease the maximum error of the original readout. The bias-corrected, *rescaled* readout, however, will now, on average, no longer match the signal. That is the effect we plot in Figure 3D.

To make these issues clearer, we now clearly specify throughout the text whether we use the readout or the corrected readout. When explaining readout and corrected readout, we explain the limits of the bounding box picture: "This bias can be largely eliminated by rescaling the readouts with a constant factor. We will sometimes use this corrected readout (see Methods, ’Readout biases’), but note that the corrected readout is not confined to stay within the bounding box."

We also added the following sentence when explaining changes in the excitability of neurons:

"At first sight, changing the box size increases or decreases the maximum error of the readout. More subtly, however, it also introduces a bias in the corrected average readout (Figure 3C, arrows)."

4. A related question applies to delays. Where exactly are the delays? In the readout and then the network is optimised for that? Or is the network the optimal one assuming no delays and then delays are added in the recurrent connections and/or the readout? Clarifying that will make it easier for the reader to follow the argument.

Crucially, the delays are in the recurrent connections. The reviewers’ second assertion is correct: Network connectivity is optimized for the delay-free case, and the delays are then added to the recurrent connections. In addition to these recurrent delays, we have also assumed a delay of identical length in the readout. This latter delay is not crucial, but facilitates the geometric visualisation, as the arrival of a delayed recurrent spike and the update of the readout thus happen at the exact same time and can thus be shown in the same figure panel.

We now clarify our choices in the Results section, where we have inserted the following statement:

"Below, we study the impact of these delays, which apply directly to recurrent excitation and inhibition. We also apply the same delays to the network readout for mathematical convenience, but those do not affect network dynamics (see Methods)."

5. In the section on perturbations of the reset potential: Is there again an asymmetry, e.g., do changes up or down in the optimal reset potential have very different consequences?

The reviewers are correct. A change in reset potential leads to a transient change in the corresponding face of the bounding box, and inward changes (a transient shrinkage of the box) can have very different effects than an outward change. Indeed, a reset that is too small can cause a successive shrinking of the box if the respective neuron fires repeatedly.

We now included the following statement in the Results section:

“We note that positive and negative changes to the default reset potential will lead to asymmetric effects on robustness like those observed for excitatory and inhibitory perturbations. Specifically, if the resets become too small, and if the leak is insufficiently fast, then successive spiking of a single neuron will draw its threshold inwards, thereby leading to a collapse of the bounding box."

Robustness claims:6. There are frequent comparisons between networks with few neurons and networks with many neurons. The idea that networks with more neurons are more robust seems kind of obvious. But the authors analyze a very specific network tuned for coordinated redundancy (Equation 3). How does that network compare to one with the same number of neurons but no coordinated redundancy? This would seem a more interesting comparison than varying the number of neurons.

We agree with the reviewer that a more systematic comparison with other types of networks would be preferable, but we have found that there are two key problems. First, there is no agreed, default network model that we could compare our framework against. Second, almost all standard network models are not built to be robust, so that systematic comparisons are not very illuminating.

We have addressed these issues in the following way:

First, we note that we include a comparison with networks without coordinated redundancy. We simply compare our networks against a set of unconnected neurons (Section ’passive redundancy’). A set of unconnected neurons is a simple example, as any perturbation effect scales linearly with the number of neurons perturbed. Here, increasing the number of neurons will not guard against the elimination of a certain fraction of the network (say 25%). In the bounding box picture, however, increasing the number of neurons (or the redundancy) will indeed increase robustness against killing one fourth of the network, as smaller networks are not necessarily robust. (We note that in both cases, we assume the same downstream readout before and after the perturbation.) This illustrates that more redundancy is not trivially better, at least not if the perturbation size scales with the network size. To highlight this better, we added the following sentence in the section "Scaling up": "This contrasts with networks of independent neurons in which performance will scale linearly with any change in redundancy for a fixed readout."

Second, we note that we illustrated the response of a trained network model without coordinated redundancy to various perturbations. This network is not robust to our cumulative perturbations, yet our identically sized network with coordinated redundancy is robust (Figure 1C vs 1D). While we only include one example simulation, we emphasize that this simulation is representative (which any reader can see by simulating our code).

Third, we would like to point out that we also show that more redundancy is not always better. Notably, our work suggests that including more neurons makes the network more sensitive to certain kinds of perturbations (e.g., synaptic noise in Figure 6G-H, or delays in Figure 7G).

7. Intuitively one might expect that the performance is fragile to changes in the recurrent coupling, since the decoding vector is tied to the feedforward and recurrent weights (Equation 10). Because of this the claimed robustness to perturbations in synaptic weight seems surprising. Is the 'synapse scaling factor' in Figure 6G,H the parameter \δ_{\Omega} in Equation 17? The way it is defined it is the maximal change, however most synapses are perturbed to a much smaller extent. Also, for \δ approaching 0.2 the performance only drops by 20% (Figure 6G). However, for such large \δ's the firing rates in the network seem unreasonably high (Figure 6H), suggesting it no longer performs efficiently. Overall it is hard to judge from the current presentation how robust the network is to synaptic perturbation. Perhaps the firing rates could be bounded, or the performance penalized for using very high rates?

The reviewer is correct that the synapse scaling factor in Figure 6G,H is the parameter *δ*_Ω_ in Equation 17, and that this corresponds to the maximal possible scaling rather than the actual scaling of the synaptic strengths.

To explain the robustness of the network against changes in the synapses, we have to first note the asymmetry of perturbations: a synapse that is smaller than its ideal value will lead to a temporary shift of the postsynaptic neuron’s threshold away from the bounding box when the presynaptic neuron spikes (Figure 6D). Just as with inhibitory perturbations, this perturbations is harmless as it does not really change the overall shape of the box in redundant systems. A synapse that is larger than its ideal value will lead to a temporary shift into the bounding box which can be harmful, depending on how long it lasts (or how fast the synaptic input decays).

We have now sought to clarify this section by writing:

"We again note an asymmetry: a synapse with decreased strength leads to an outward move of the postsynaptic neuron’s threshold, which is generally harmless. Random synaptic failures, which cause temporary decreases in synaptic strength, do therefore not influence the bounding box functionality. However, a synapse with increased strength leads to an inward move, which could be a temporarily harmful perturbation."

We also agree with the reviewer that synaptic scaling can make these networks less inefficient in terms of number of spikes (Figure 6H). We now write:

"Overall, we find that more redundant networks (with consequently more synapses) are typically more vulnerable to these perturbations, and that synaptic scaling can lead to highly inefficient networks in terms of spike rate, regardless of the network redundancy."

Relation to plasticity:8. Prior work (e.g. Bourdoukan 2012) proposed a Hebbian learning process that could tune up such a network automatically. If this plasticity is "always on" how would it respond to the various perturbations being considered? How might that contribute to robustness on a longer time scale? How does the size/shape of the bounding box depend on the parameters of the learning rule? Can one get looser and tighter boxes, as needed here for certain types of robustness?

The learning rules we have previously developed (Bourdoukan et al., 2012, Neurips; Brendel et al., 2020, PLOS CB), will learn an optimally arranged bounding box with a pre-defined width. We have not systematically studied how perturbations would affect the learning—a very interesting question, but we believe beyond the confines of the current study.

Still, a few answers we already know. Specifically, perturbations that lower the excitability of neurons are unlikely to affect learning. For instance, when eliminating neurons, the target connectivity of the learning rules does not change. Moreover, the learning rules we developed will still push synapses in the remaining neurons towards this target connectivity. In machine-learning language, the networks are fully capable of dealing with the dropout of neurons, and dropout may even help in speeding up learning (although we have not full studied this effect). More generally, inhibiting neurons does not affect learning because of the network’s compensatory properties. Perturbations that excite neurons (e.g. synaptic noise) are different. We would speculate that they are likely to strongly perturb the learning process.